# Concerted conformational dynamics and water movements in the ghrelin G protein-coupled receptor

Maxime Louet[1], Marina Casiraghi[2†‡], Marjorie Damian[1], Mauricio GS Costa[3,4], Pedro Renault[1§], Antoniel AS Gomes[1,5], Paulo R Batista[4], Céline M'Kadmi[1], Sophie Mary[1], Sonia Cantel[1], Severine Denoyelle[1], Khoubaib Ben Haj Salah[1], David Perahia[3], Paulo M Bisch[5], Jean-Alain Fehrentz[1], Laurent J Catoire[2], Nicolas Floquet[1], Jean-Louis Banères[1]*

[1]IBMM, Univ Montpellier, CNRS, ENSCM, Montpellier, France; [2]Laboratoire de Biologie Physico-Chimique des Protéines Membranaires, UMR 7099, CNRS, Université de Paris, Institut de Biologie Physico-Chimique (FRC 550), Paris, France; [3]Laboratoire de Biologie et Pharmacologie Appliquées, UMR 8113 CNRS, Ecole Normale Supérieure Paris-Saclay, Gif-sur-Yvette, France; [4]Programa de Computação Científica, Fundação Oswaldo Cruz, Rio de Janeiro, Brazil; [5]Laboratório de Física Biológica, Instituto de Biofísica Carlos Chagas Filho, Universidade Federal do Rio de Janeiro, Rio de Janeiro, Brazil

**\*For correspondence:**
jean-louis.baneres@umontpellier.fr

**Present address:** †Department of Molecular and Cellular Physiology, Stanford University School of Medicine, Stanford, United States; ‡IBMM, Univ Montpellier, CNRS, ENSCM, Montpellier, France; §Laboratory of Molecular Neuropharmacology and Bioinformatics, Institut de Neurociències, Universitat Autònoma de Barcelona, Barcelona, Spain

**Competing interests:** The authors declare that no competing interests exist.

**Abstract** There is increasing support for water molecules playing a role in signal propagation through G protein-coupled receptors (GPCRs). However, exploration of the hydration features of GPCRs is still in its infancy. Here, we combined site-specific labeling with unnatural amino acids to molecular dynamics to delineate how local hydration of the ghrelin receptor growth hormone secretagogue receptor (GHSR) is rearranged upon activation. We found that GHSR is characterized by a specific hydration pattern that is selectively remodeled by pharmacologically distinct ligands and by the lipid environment. This process is directly related to the concerted movements of the transmembrane domains of the receptor. These results demonstrate that the conformational dynamics of GHSR are tightly coupled to the movements of internal water molecules, further enhancing our understanding of the molecular bases of GPCR-mediated signaling.

## Introduction

G protein-coupled receptors (GPCRs) are major players in many central biological processes (*Lagerström and Schiöth, 2008*). The diversity in the signaling properties of GPCRs indicates that this process cannot be fully described by the limited number of conformational states captured by X-ray crystallography and cryoelectron microscopy (cryo-EM). Indeed, GPCRs likely explore complex conformational landscapes, characterized by several meta-stable structural states. The relative distribution of these states is controlled by ligands, signaling proteins, and the environment, ultimately dictating the signaling output (*Casiraghi et al., 2019*; *Hilger et al., 2018*; *Wingler and Lefkowitz, 2020*). As a consequence, the conformational dynamics of GPCRs and its modulation by the receptor's environment are under intense scrutiny, as this should illuminate how signal transduction occurs.

Among all the components in the receptor environment that control receptor dynamics and signaling behavior, the solvent is to play an important but yet unexplored role. Many GPCR experimental structures indicate the occurrence of water molecules within their transmembrane (TM) regions.

Some are located in the ligand-binding pocket and directly contribute to the energetics of ligand binding (*Deflorian et al., 2020*). Others lie in different cavities and have been proposed to be central to the allosteric propagation of the conformational rearrangements required for receptor activation (*Lesca et al., 2018*; *Huang et al., 2015*; *Zhao et al., 2020*). Yet, mechanistic models describing the dynamics and role of water molecules in GPCR functioning remain speculative, as only limited experimental information is available. Indeed, the relationship between GPCR conformational dynamics and local hydration has been mostly inferred from molecular dynamics (MD) simulations with rhodopsin (*Grossfield et al., 2008*) and other receptors as well (*Yuan et al., 2014*; *Venkatakrishnan et al., 2019*). Although these simulations provide invaluable information on the arrangement and movements of water molecules, they nevertheless require further experimental support.

To analyze the hydration pattern of GPCRs, we used here an original strategy initially described with a model soluble protein (*Amaro et al., 2015*). This strategy combines site-specific labeling with unnatural amino acids (UAAs), fluorescence spectroscopy, and MD. This approach was applied to the growth hormone secretagogue receptor (GHSR). In addition to being a model for class A GPCRs, GHSR is a major target in pharmacology. Indeed, this receptor and its natural peptide agonist, ghrelin, are involved in most important biological processes such as the control of food intake, glucose metabolism, or reward and stress behaviors (*Müller et al., 2015*). Using the emission properties of a particular UAA whose fluorescence emission properties are related to its hydration (*Amaro et al., 2015*; *Choudhury et al., 2008*), we found here that the ghrelin receptor hydration pattern is likely remodeled by orthosteric ligands and the lipid environment. In parallel, MD simulations provided a structural framework to the fluorescence observations and demonstrated that such a remodeling may be associated with collective movements of GHSR TM domains. Taken together, our data illuminate GPCR signaling with a mechanism where specific changes in the local hydration of the receptor could occur in a concerted manner with its conformational dynamics, in direct relationship to the activation process.

## Results

### GHSR labeling

We used L-(7-hydroxycoumarin-4-yl)-ethylglycine as a reporter of receptor local hydration. This UAA contains the L-(7-hydroxycoumarin-4-yl) (7H4MC) moiety whose emission properties are correlated to the presence of water molecules in its vicinity (*Amaro et al., 2015*). 7H4MC-ethylglycine was synthesized as described in the Materials and methods section and introduced in GHSR using codon suppression technology (*Wang et al., 2006*). To analyze receptor activation in a relevant membrane-like environment, the labeled receptor was inserted into lipid nanodiscs formed by the scaffolding MSP1E3D1 protein and a POPC:POPG mixture (see Materials and methods) (*Damian et al., 2012*). Under such conditions, homogeneous nanodisc populations of functional receptors were obtained (*Figure 1—figure supplement 1*). Of importance, the active receptor was purified through a ligand affinity chromatography step to ensure all the receptors in our preparations were competent with regard to ligand binding (*Ferré et al., 2019*). Accordingly, we repeatedly demonstrated that the receptor obtained under such conditions is totally functional with regard to ligand binding, and that its pharmacological profile is closely related to that of GHSR expressed in HEK cell membranes (*Ferré et al., 2019*).

7H4MC-ethylglycine was introduced at several, single positions within the TM domains of GHSR, namely Y81$^{2.42}$, W104$^{2.65}$, Y106$^{2.67}$, F119$^{3.28}$, F121$^{3.30}$, I134$^{3.43}$, F179$^{4.61}$, W215$^{5.41}$, Y232$^{5.58}$, V268$^{6.40}$, F272$^{6.44}$, Y303$^{7.33}$, or S315$^{7.45}$ (superscript numbers follow Ballesteros-Weinstein numbering [*Ballesteros and Weinstein, 1995*; *Figure 1A*]). In all the cases, protein expression yields markedly decreased but, with the exception of the F119$^{3.28}$ mutant that was not expressed at a detectable level, the amounts of purified receptor obtained were still compatible with the fluorescence experiments, that is, in the range of a hundred of µg per liter of bacterial culture. However, the modified receptors bearing 7H4MC-ethylglycine at position Y81$^{2.42}$, W104$^{2.65}$, Y106$^{2.67}$, F119$^{3.28}$, F121$^{3.30}$, F179$^{4.61}$, and Y303$^{7.43}$ could not be purified through the ligand affinity chromatography step, indicating that replacing the native residue with 7H4MC-ethylglycine affected their three-dimensional fold and/or their ability to bind ligands. In addition, replacing W215$^{5.41}$ with

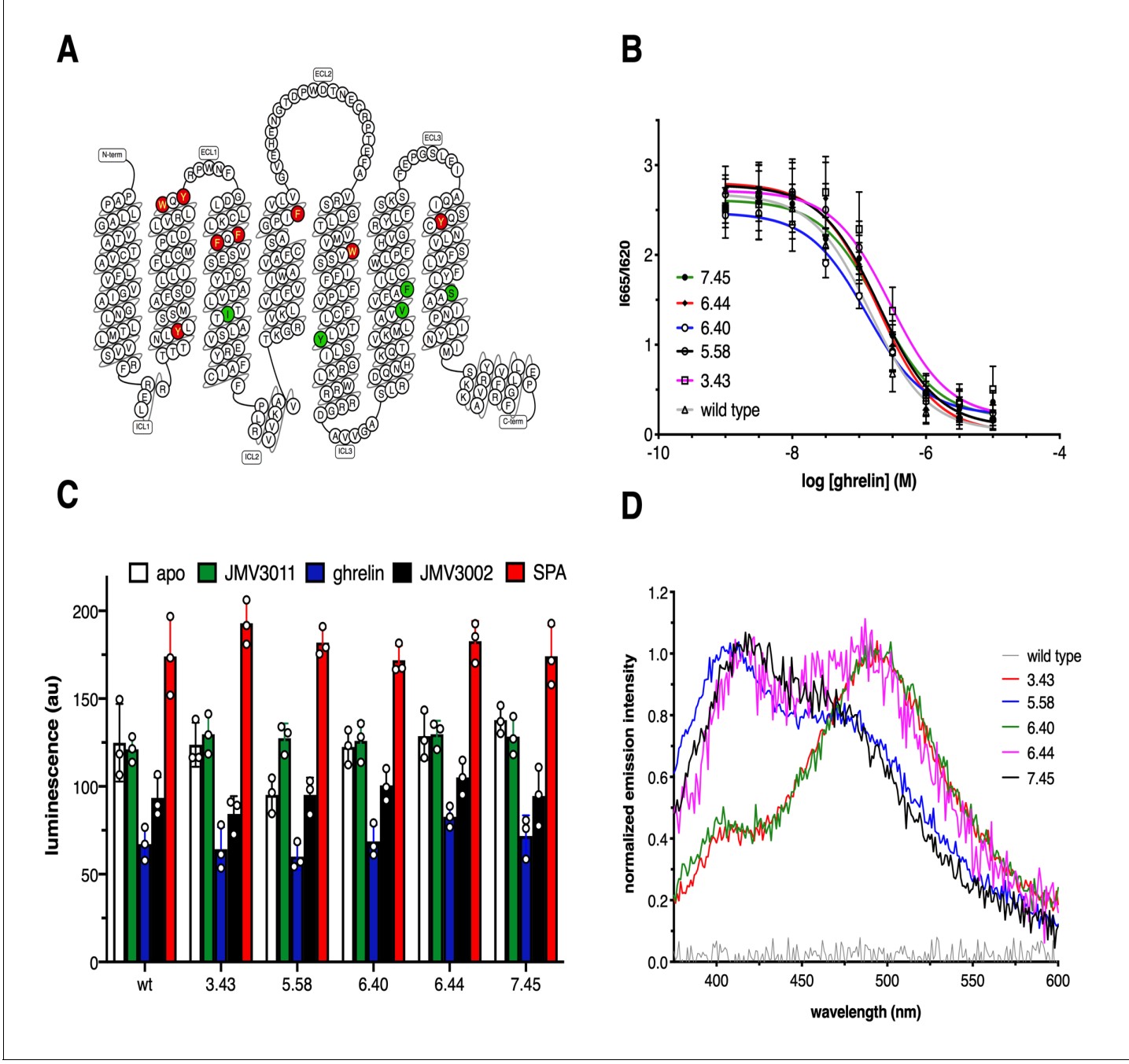

**Figure 1.** Growth hormone secretagogue receptor (GHSR) labeling. (A) Position of the labeled residues in GHSR sequence. Red labeling indicates positions that were deleterious to GHSR expression and/or function. Green labeling indicates positions that did not markedly affect the pharmacological properties of the isolated receptor and were considered in the present work. (B) FRET-monitored competition assays of ghrelin for binding to GHSR assembled into nanodiscs. (C) GTP turnover for Gq catalyzed by GHSR and its labeled counterparts in the absence of ligand (apo) or in the presence of 10 μM of JMV3011, ghrelin, JMV3002, or SPA (substance-P analog). (D) Normalized emission spectrum of the apo wild-type and labeled GHSR with $\lambda_{exc}$ set at 320 nm. Data in (B) and (C) is the mean value ± SD of three experiments. Statistical analyses for the data in (C) are provided in *Figure 1—figure supplement 3*.

The online version of this article includes the following source data and figure supplement(s) for figure 1:

**Source data 1.** HTRF ratio for GHSR and its mutants.

**Source data 2.** Luminescence values for the GTP turnover assay.

**Source data 3.** GHSR emission intensity.

**Figure supplement 1.** The growth hormone secretagogue receptor (GHSR)-containing nanodiscs.

*Figure 1 continued on next page*

*Figure 1 continued*

**Figure supplement 1—source data 1.** SDS-PAGE of the GHSR-containing nanodiscs.
**Figure supplement 2.** Control for the ligand-binding properties of the isolated growth hormone secretagogue receptor (GHSR) in nanodiscs.
**Figure supplement 2—source data 1.** HTRF ratio for GHSR- and BLT-containing discs.
**Figure supplement 3.** Statistical analysis of the GTP turnover assay.
**Figure supplement 4.** N-terminal labeling of growth hormone secretagogue receptor (GHSR).
**Figure supplement 4—source data 1.** Absorbance of GHSR-containing nanodiscs before and after TEV cleavage.
**Figure supplement 4—source data 2.** Emission intensity of GHSR-containing nanodiscs before and after TEV cleavage.
**Figure supplement 5.** Fluorescent ghrelin characterization.
**Figure supplement 6.** The G protein trimer.
**Figure supplement 6—source data 1.** SDS-PAGE profile of the G protein used in the GTP-binding assays.

7H4MC-ethylglycine markedly decreased the basal activity of the receptor, although this mutant could still bind its ligands and be activated by ghrelin. In contrast, for the other positions, that is, I134$^{3.43}$, Y232$^{5.58}$, V268$^{6.40}$, F272$^{6.44}$, and S315$^{7.45}$, replacing the naturally occurring residue with 7H4MC-ethylglycine affected neither ghrelin binding nor the receptor-catalyzed Gq activation in a relevant manner (*Figure 1B, C*, *Figure 1—figure supplement 2*, *3*, *4*, *5*, *6*). Hence, only these mutants were considered in our analyses. As shown in *Figure 1D*, the modified proteins displayed an emission spectrum characteristic of the 7H4MC moiety, while the wild-type receptor had no significant emission signal when excited at the same wavelength. This indicates an efficient incorporation of the labeled UAA into the receptor.

## GHSR local hydration

We then investigated whether the fluorescence properties of 7H4MC-ethylglycine could report on the local hydration features of GHSR. To this end, we analyzed the 7H4MC emission profile for each of the positions considered. An excitation wavelength of 320 nm was systematically used to excite the neutral form of the fluorophore (*Amaro et al., 2015*). The emission spectra were deconvoluted into their separate components using the procedure initially described (*Amaro et al., 2015*). A hydration parameter H was then determined that corresponded to the sum of the contributions of the anionic and tautomer forms. This parameter is an indicator of the extent of hydration at the position considered, as the higher the H parameter the higher local hydration (*Amaro et al., 2015*). A difference in the H parameter inferred for the 7H4MC probe at the different positions of the apo GHSR was observed depending on the position considered (*Figure 2—figure supplement 1*, *2*). Indeed, all positions were hydrated to some extent, but some displayed a high H value characteristic of high hydration (3.43, 6.40) whereas others displayed a low H value suggestive of a lower local hydration (5.58, 6.44, 7.45). This indicates that 7H4MC fluorescence is a good indicator to discriminate between different local hydration states in the receptor structure. Besides, these data show that local hydration, as reported by 7H4MC fluorescence, depends on the region of the TM domain considered, with some regions more accessible to the solvent than others, even for closely related positions in the receptor structure (e.g., V268$^{6.40}$ and F272$^{6.44}$).

## Impact of ligands on GHSR local hydration

We then used 7H4MC fluorescence to monitor the impact of ligand binding on the hydration pattern of GHSR. To this end, the H parameter was measured in the presence of saturating concentrations in ligands from different pharmacological classes, that is, the natural full agonist (ghrelin), a neutral antagonist (JMV3011), a Gq-biased partial agonist (JMV3002), and an inverse agonist (substance-P analog [SPA]) (*M'Kadmi et al., 2015*; *Figure 2—figure supplement 3*). Binding of JMV3011 to labeled GHSR was not accompanied by a measurable change in the hydration parameter for any of the positions considered (*Figure 2*, *Figure 2—figure supplement 4*). This is to be related to our previous observations demonstrating that binding of this compound was not associated with any change in the conformation of isolated GHSR (*Mary et al., 2012*; *Damian et al., 2015*). In contrast, changes in the H parameter were observed at some specific positions upon binding of either the full agonist ghrelin, the Gq-biased agonist JMV3002, or the inverse agonist SPA. Specifically, ghrelin binding was associated with an increase in the hydration parameter at position 5.58 while local

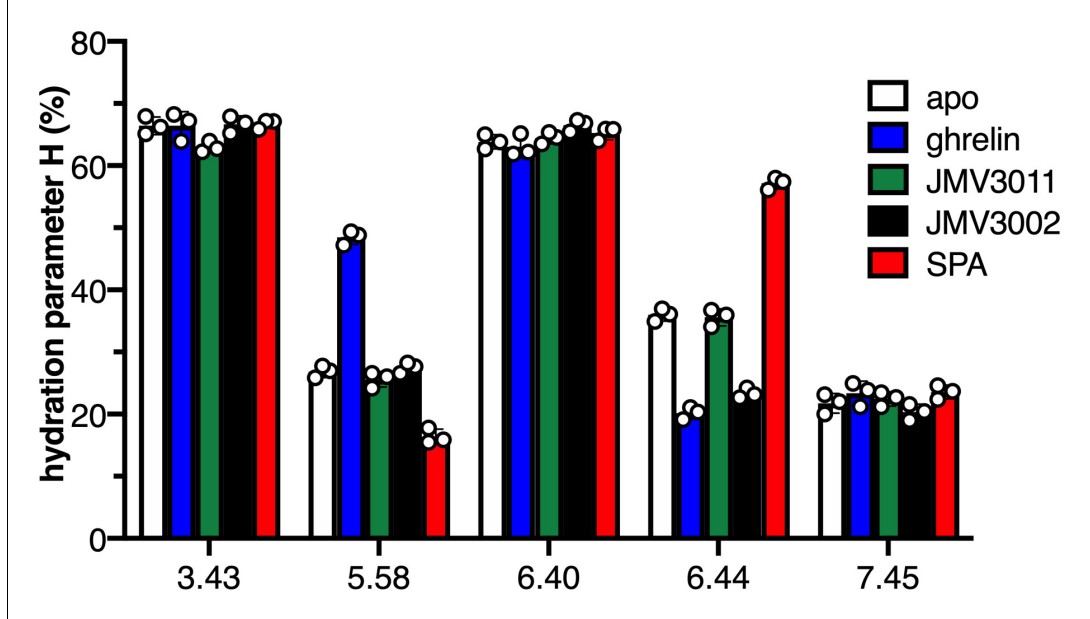

**Figure 2.** Local hydration of growth hormone secretagogue receptor (GHSR) as a function of ligands. H parameter for the 7H4MC-labeled GHSR in the absence of ligand (apo) and in the presence of JMV3011, ghrelin, JMV3002, or SPA (substance-P analog). All ligands were used at a 10 μM concentration. In all cases, the data represents the mean value ± SD of three experiments. Statistical analyses are provided in *Figure 1—figure supplement 2* and *4*.

The online version of this article includes the following source data and figure supplement(s) for figure 2:

**Source data 1.** H parameter for GHSR and its mutants.

**Figure supplement 1.** Emission spectra of L-(7-hydroxycoumarin-4-yl) (7H4MC)-ethylglycine incorporated into isolated growth hormone secretagogue receptor (GHSR).

**Figure supplement 1—source data 1.** 7-H4MC emission intensity of the apo GHSR mutants.

**Figure supplement 2.** Statistical analysis of the data in *Figure 2*.

**Figure supplement 3.** Structure of the ligands used throughout this work.

**Figure supplement 4.** Statistical analysis of the data in *Figure 2*.

**Figure supplement 5.** Emission spectra of L-(7-hydroxycoumarin-4-yl) (7H4MC)-ethylglycine incorporated at position 5.58 of growth hormone secretagogue receptor (GHSR) in the presence of pharmacologically distinct ligands or phosphatidylinositol-4,5-bisphosphate (PIP2).

**Figure supplement 5—source data 1.** 7-H4MC emission intensity of the GHSR 5.58 mutant in the presence of ligands.

**Figure supplement 6.** Emission spectra of L-(7-hydroxycoumarin-4-yl) (7H4MC)-ethylglycine incorporated at position 6.44 of growth hormone secretagogue receptor (GHSR) in the presence of pharmacologically distinct ligands or phosphatidylinositol-4,5-bisphosphate (PIP2) in the nanodiscs.

**Figure supplement 6—source data 1.** pa 7-H4MC emission intensity of the GHSR 6.44 mutant in the presence of ligands.

hydration at position 6.44 decreased (*Figure 2—figure supplement 4*, *5*, *6*). Besides 5.58 and 6.44, no relevant change in the hydration parameter was observed for the other positions (*Figure 2—figure supplement 4*). This indicates that agonist-induced GHSR activation is accompanied by a concerted parallel increase and decrease of the local hydration in specific regions of the receptor, namely here TM5 and TM6. Interestingly, no change was observed for V268[6.40] whereas a decrease in the H parameter was measured for the probe at F272[6.44], although both positions are close in GHSR structure. This suggests that 7H4MC fluorescence is well adapted to monitor hydration changes in a very local environment and that changes in local hydration likely occur at specific, spatially restricted sites. Perhaps not surprisingly, binding of the inverse agonist SPA was accompanied by a change in the hydration parameter opposite to that observed with ghrelin, that is, the H parameter decreased and increased for positions 5.58 and 6.44, respectively (*Figure 2*, *Figure 2—figure supplement 4*), consistent with the opposite effect of ghrelin and SPA on GHSR activation and conformational landscape (*Mary et al., 2012*). Finally, the hydration pattern in the presence of JMV3002 was different from that observed in the presence of ghrelin. Indeed, while the binding of this compound was still accompanied by a decrease in the hydration parameter at position 6.44, as in the case of ghrelin binding, no change was observed for the probe at position 5.58 (*Figure 2*, *Figure 2—*

*figure supplement 4*). This could be related to the differences in the pharmacological profile of the two compounds, as ghrelin is a full agonist whereas JMV3002 is a Gq partial agonist that triggers neither Gi activation nor arrestin recruitment (*M'Kadmi et al., 2015*).

## Structural bases of the changes in water accessibility

To provide a structural framework to our experimental observations and observe possible differences between the hydration pattern of inactive and active-like conformers of wild-type GHSR, we then ran five MD simulations of 5 µs for each conformational state of the receptor, totalizing 50 µs of all-atoms simulation. The crystal structure of the inactive, antagonist-loaded state of the receptor has been solved (*Shiimura et al., 2020*) and was used as a starting point for our MD studies. Besides, two cryo-EM structures of the ghrelin receptor in complex with ghrelin or a synthetic agonist and a Gq mimetic have been posted on the BioRxiv preprint server (https://doi.org/10.1101/2021.06.09.447478). However, since the coordinates of these structures are not yet available, we had to model an active-like state of GHSR in the absence of its cognate G protein (see Materials and methods). A projection of all conformers explored during our simulations confirmed their compatibility with experimental structures, describing mainly inactive and intermediate states (based on the classification in the GPCRdb; *Pándy-Szekeres et al., 2018*), the latter corresponding to an activated receptor without the G protein (*Figure 3—figure supplement 1*).

Interestingly, analysis of the statistical water distribution in GHSR confirmed that differences in the hydration pattern could exist depending on its inactive/active states (*Figure 3*). More importantly, these differences effectively occurred in the regions where the 7H4MC-ethylglycine residue had been inserted in our experiments. Four out of five simulations starting from the X-ray (inactive) structure of GHSR converged toward a same hydration pattern (*Figure 3A-D*). In the last simulation (*Figure 3E*), the water statistically occupied a larger volume on the intracellular side of the receptor. Of interest, this distribution of water molecules in the inactive state was in agreement with the

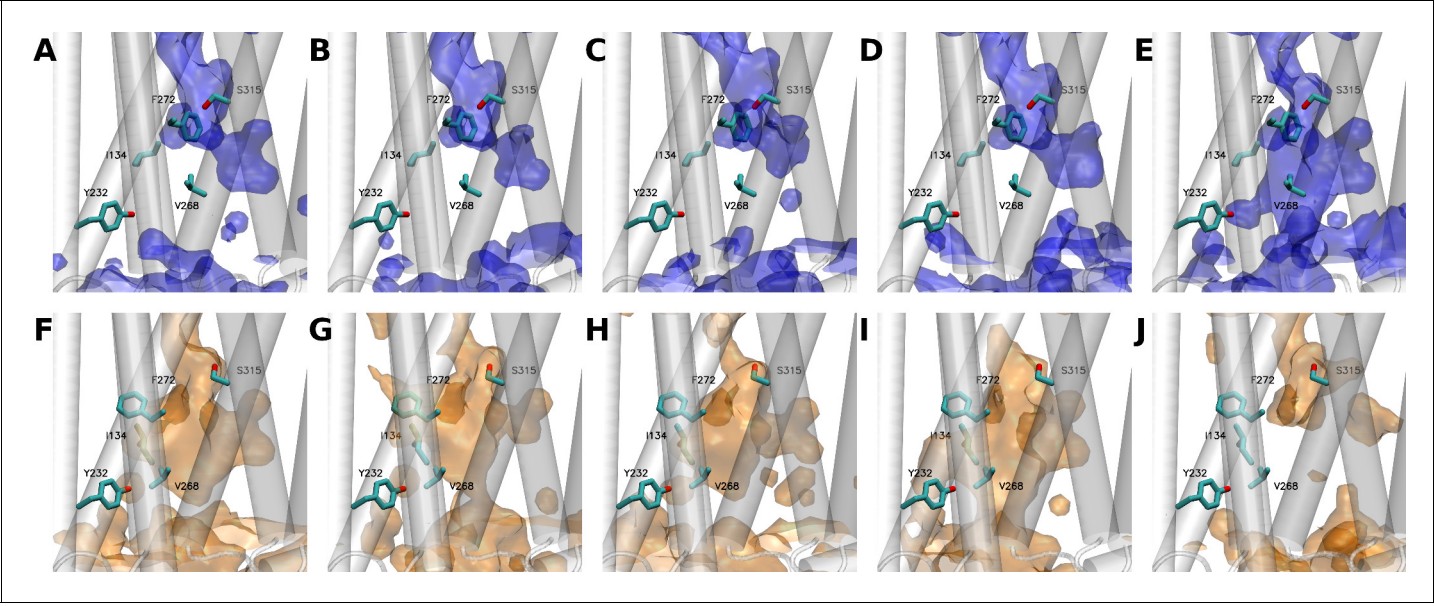

**Figure 3.** Water distribution in growth hormone secretagogue receptor (GHSR) as observed along the five independent 5 µs molecular dynamics (MD) simulations starting from either the inactive (A to E) or the active (F to J) states of the receptor. (E and J) panels show simulations where GHSR transited from inactive to active (E) or from active to inactive (J) states, respectively. The backbone of the protein is represented as a transparent-white cartoon, while the five positions at which the L-(7-hydroxycoumarin-4-yl) (7H4MC)-ethylglycine residue was inserted are shown in licorice. Blue or orange surfaces respectively describe the hydration of the receptor using a probability of 0.3. Volumetric maps were computed using the volmap tool of visual molecular dynamics (VMD).

The online version of this article includes the following figure supplement(s) for figure 3:

**Figure supplement 1.** Projection of structures and molecular dynamics (MD) conformers on the first two eigenvectors obtained from the principal component analysis (PCA) of experimental structures.

distribution described recently for other receptors of the same family using similar methods (*Venkatakrishnan et al., 2019*; *Bertalan et al., 2020*). In agreement with our experimental data, the water distribution in the inactive conformation of the receptor showed the presence of water molecules around F272$^{6.44}$ whereas Y232$^{5.58}$ was not solvated. Accordingly, I134$^{3.43}$ was also in contact with water molecules. However, and in contradiction with our experiments, V268$^{6.40}$ was not in contact with water molecule in the inactive state whereas S315$^{7.45}$ was. V268$^{6.40}$ occupies a central position in the receptor whereas S315$^{7.45}$ is close to the interface between TM6 and TM7 (*Figure 4*). If insertion of 7H4MC-ethylglycine at positions 134$^{3.43}$, 232$^{5.58}$, and 272$^{6.44}$ are more conservative in terms of residue size, the insertion of 7H4MC in place of a valine (V268$^{6.40}$) or a serine (S315$^{7.45}$) suggests a stronger adaptation of the receptor fold to these mutations. To clarify the possible orientations of 7H4MC-ethylglycine into the receptor, we thus computed adiabatic maps for positions 268$^{6.40}$ and 315$^{7.45}$ (*Figure 4—figure supplement 1*). In the case of 7H4MC-ethylglycine at position 268$^{6.40}$, the adiabatic map confirmed that this large residue, in comparison to a valine, allowed interaction with solvent molecules (*Figure 4—figure supplement 1*). For the 315$^{7.45}$ position, even the adiabatic map suggested that this residue could conserve its initial orientation toward the interior of the receptor and should be highly solvated.

Interestingly, a different hydration pattern was found in the fifth simulation. This profile was explained by a spreading of TM6 during the simulation, thus leading to conformers close to those observed when starting from the active-like state (*Figure 4—figure supplement 2*). Accordingly, the resulting hydration pattern was very close to that obtained in the simulations starting from the active-like state (*Figure 3F-I*). In this pattern, water molecules were more uniformly distributed in the receptor including its lower, intracellular part. Indeed, the main structural difference between both states was the spreading of TM6 (*Figure 4—figure supplement 2*), which contributed to a large water influx into the intracellular moiety of the receptor. In this pattern, and in agreement with the fluorescence experiments, I134$^{3.43}$, Y232$^{6.44}$, and V268$^{6.40}$ were all in close contact with water molecules whereas F272$^{6.44}$ was flipped toward TM5, contributing to reduce its interactions with

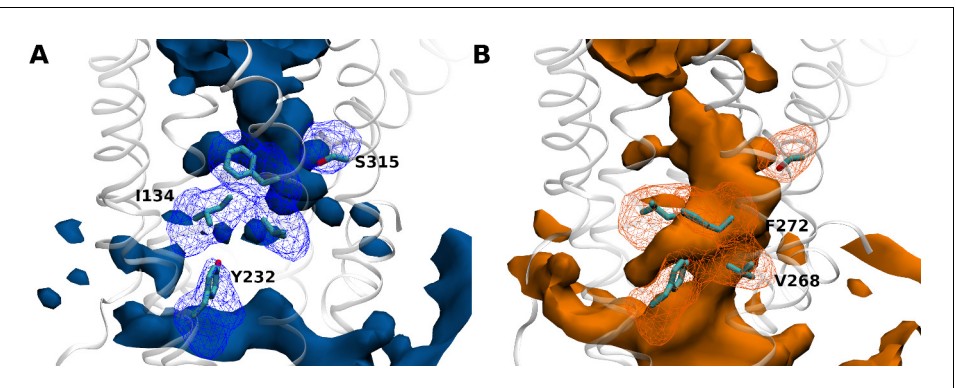

**Figure 4.** Amino acid positions and hydration patterns of inactive growth hormone secretagogue receptor (GHSR) (**A**) and active GHSR (**B**) explored by molecular dynamics (MD) simulations. GHSR is represented in white ribbons. Volumetric maps in solid surface represent the water distribution with a probability of presence of 0.3. Meshes represent the most probable (probability of 0.3) positions of residues I134$^{3.43}$, Y232$^{5.58}$, V268$^{6.40}$, F272$^{6.44}$, and S315$^{7.45}$ in both states. Snapshots representing the residues in their respective shapes are drawn in licorice for visualization. Volumetric maps were computed using the volmap tool of visual molecular dynamics (VMD).

The online version of this article includes the following figure supplement(s) for figure 4:

**Figure supplement 1.** Intersections between water distributions and predicted L-(7-hydroxycoumarin-4-yl) (7H4MC) orientations in the inactive growth hormone secretagogue receptor (GHSR) (**A and B**) and the active GHSR (**C and D**) at positions 268 (**A and C**) and 315 (**B and D**).

**Figure supplement 2.** Structural differences between both starting conformers: inactive (blue) and active (orange).

**Figure supplement 3.** Orientation of F272$^{6.44}$ in all experimental-inactive (**A**) and experimental-active (**B**) structures.

**Figure supplement 4.** Motions encoded by the first two eigenvectors resulting from the principal component analysis (PCA) of the conserved Cα coordinates from all experimental structures.

**Figure supplement 5.** Residue selection for principal component analysis (PCA).

surrounding water (*Figure 4—figure supplement 3*). This structural feature was found to be conserved in all GPCRs (*Figure 4*).

We also obtained a simulation starting from the active state that showed a different behavior from other simulations, that is, a closure and a loss of hydration in the intracellular part of the receptor due to a motion of TM7 inside the receptor (*Figure 3J*). Such a motion was compatible with the direction coded by the experimental structures and shown by principal component analysis (PCA), where the first two eigenvectors displayed this inward motion of TM7 concomitant to TM6 spreading (*Figure 4—figure supplement 4*). Accordingly, in this simulation, the resulting hydration pattern was very close to those observed in the simulations starting from the inactive state (*Figure 3A-D*).

## Impact of lipids on GHSR local hydration

In their native environment, receptors are surrounded not only by the solvent but also by the lipid bilayer. To provide an illustration of the impact of the environment of the ghrelin receptor on its local hydration, we finally analyzed the effect of the lipid composition of the nanodiscs on 7H4MC fluorescence for the two positions that were affected by receptor activation. Specifically, we measured the hydration parameter for the probe at positions 5.58 and 6.44 with GHSR assembled into POPC:POPG nanodiscs in the absence or presence of phosphatidylinositol-4,5-bisphosphate (PIP2), a lipid that has been shown to impact on the activity of many different membrane proteins (*Hammond and Burke, 2020*) including GPCRs (*Yen et al., 2018*) and, more recently, the ghrelin

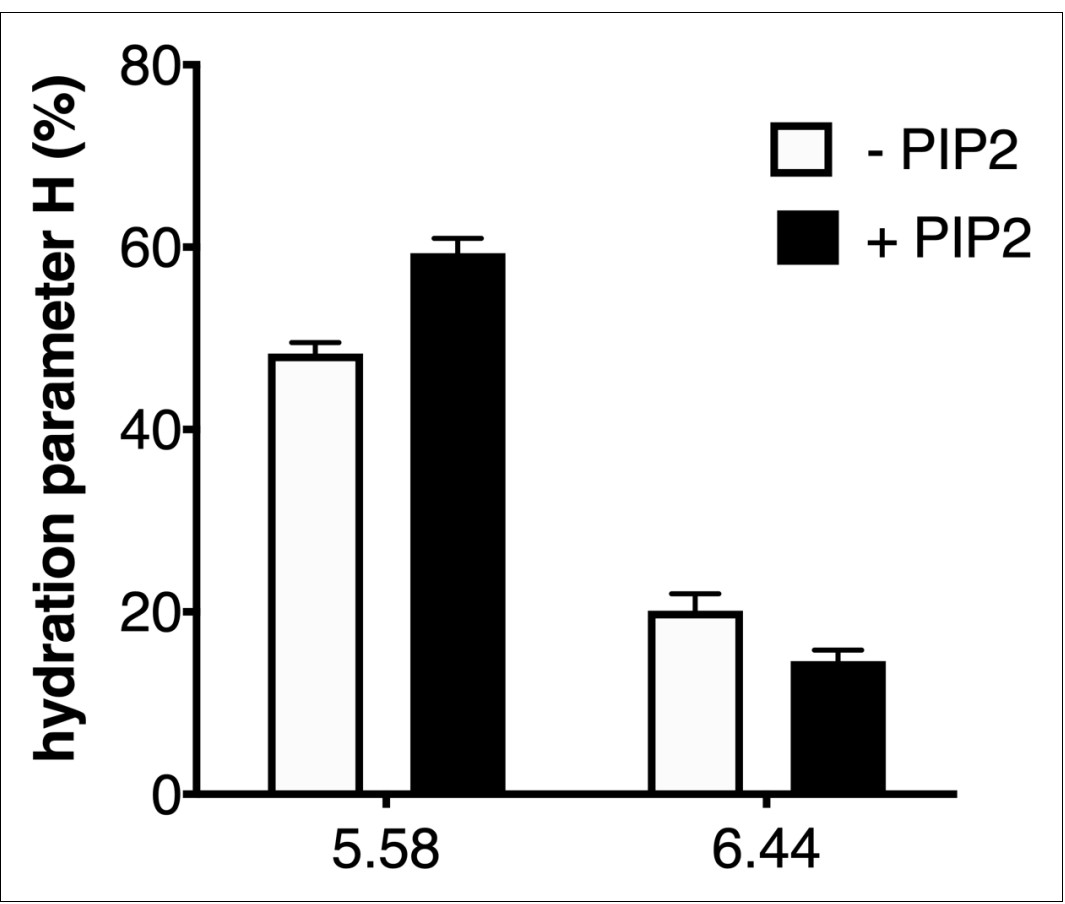

**Figure 5.** Impact of lipids on the local hydration of growth hormone secretagogue receptor (GHSR). H parameter for L-(7-hydroxycoumarin-4-yl) (7H4MC)-labeled GHSR assembled into nanodiscs containing or not phosphatidylinositol-4,5-bisphosphate (PIP2) (2.5% PIP2-to-total lipids molar ratio), in the presence of 10 µM ghrelin. The data represents the mean value ± SD of three experiments.

The online version of this article includes the following source data for figure 5:

**Source data 1.** H parameter as a function of PIP2 in the nanodiscs.

receptor (*Damian et al., 2021*). As shown in *Figure 5*, the H parameter for the two positions that were sensitive to receptor activation was further affected by PIP2. Indeed, adding 2.5% of this lipid to the nanodiscs increased the H parameter for position 5.58 and concomitantly decreased that for position 6.44. PIP2 therefore exalted the effect of the full agonist ghrelin had on the receptor hydration features. This suggests that PIP2 further shifts the conformational equilibrium toward hydration pattern associated with the active-like conformation of GHSR, indicative of an allosteric coupling between the full agonist and PIP2 for stabilizing this state. This effect could explain the impact of PIP2 on GHSR-catalyzed G protein activation (*Damian et al., 2021*).

## Discussion

Local hydration and polar networks have been proposed to play a role in the allosteric activation process of GPCRs, as it is the case for many other proteins (*Leitner et al., 2020*). However, illuminating this role is complicated by the lack of straightforward experimental approaches that could help delineate the hydration pattern of receptors under a variety of conditions. To decipher the concerted changes in the hydration pattern and conformational repertoire of GHSR, we used here a strategy combining advanced MD to fluorescence spectroscopy, an approach that had been previously developed with a model protein (*Amaro et al., 2015*). Specifically, 7H4MC-ethylglycine was incorporated at specific positions in GHSR through codon suppression technology. 7H4MC-ethylglycine includes the 7H4MC fluorophore whose emission properties are directly dependent on the water content in its microenvironment (*Georgieva et al., 2005*). Of importance, this chromophore has been shown not to affect the global hydration levels but, in some particular situations, only to influence the residence time of water molecules (*Georgieva et al., 2005*). It therefore should not markedly modify the hydration pattern of the protein, with the condition it does not affect its three-dimensional fold (*Georgieva et al., 2005*).

The fluorescent probe was introduced at different positions along GHSR sequence. Among all the mutants we considered, only those involving I134$^{3.43}$, Y232$^{5.58}$, V268$^{6.40}$, F272$^{6.44}$, and S315$^{7.45}$ were essentially neutral with regard to the ability of the receptor to bind ghrelin and activate Gq. This suggests that incorporation of 7H4MC-ethylglycine at these positions does not dramatically perturb the three-dimensional fold of the ghrelin receptor. The absence of major effect on substituting F272$^{6.44}$ with 7H4MC-ethylglycine on GHSR functioning was puzzling, as mutating this residue to an alanine had been shown to abolish GHSR constitutive activity and reduce ghrelin-induced signaling without affecting ghrelin binding affinity (*Valentin-Hansen et al., 2012*). However, replacing the phenylalanine with a tyrosine had a far lower impact on both basal- and ghrelin-induced GHSR activity (*Valentin-Hansen et al., 2012*). As concluded by the authors, a possible mechanism would be that a rigid, aromatic group is required at this position for stabilizing the receptor active state. This could explain why replacing phenylalanine with 7H4MC-ethylglycine did not lead to a major decrease in GHSR activity.

The fluorescence properties of 7H4MC were affected by the activation state of GHSR only when the probe was located at positions 5.58 and 6.44. Besides these two positions, highly hydrated, invariant positions were observed. These could correspond to regions of the receptor directly accessible to the solvent whatever its activation state is. Alternatively, hydration at these positions could include the contribution of water molecules with a structural rather than a functional role. Changes in the hydration pattern of GPCRs have been shown either to involve a direct rearrangement of the water molecules in the receptor ligand-binding pocket upon ligand binding or to result from the conformational changes associated with receptor activation. The two positions where we observed a change in the H parameter upon agonist binding, that is, Y232$^{5.58}$ and F272$^{6.44}$, are both located outside the major putative ligand-binding pocket of GHSR (*Shiimura et al., 2020*). The effects we observed on the H parameter upon ligand binding are thus not likely to be the direct consequence of the rearrangement of water molecules within the GHSR binding pocket but may rather result from differences in the hydration pattern of the different conformational states of GHSR, as fully supported by our MD simulations.

The emission profile of 7H4MC reflects the average, equilibrium contribution of the hydration pattern in the different GHSR states present in the solution. Any change in the emission properties of the probe thus implies (i) a change in the distribution of the different states in the receptor conformational landscape and (ii) a difference in the hydration features of these states. Taken together, the

variations we observed therefore demonstrate that the remodeling of the hydration pattern is an integral component of the rearrangement of the GHSR conformational repertoire associated with receptor activation. This remodeling is directly correlated to concerted, specific and well-defined movements in the TM domains of the receptor, as demonstrated by MD. This is fully consistent with previous work demonstrating the role of water molecules in the activation process of other GPCRs such as the GLP-1 receptor (*Zhao et al., 2020*; *Wootten et al., 2016a*; *Wootten et al., 2016b*). Whether our data reflects the fact that water molecules are allosteric players in the activation process, as demonstrated for other proteins including rhodopsin (*Chawla et al., 2021*), or that the movements of water molecules compensate the changes in the intramolecular voids within the different states involved in receptor activation remains an open question.

In parallel to the fluorescence approach, we used a complementary approach based on MD simulations that was aimed at providing a structural framework to the experimental observables. This method is dedicated to the exploration of the conformational space accessible to the receptor without any bias. Indeed, structures from the Protein Data Bank (PDB) can help in identifying hotspots for protein-water interactions, but in more than half of the available structures the resolution is not good enough to observe any water molecules. Moreover, static structures do not inform about the dynamical behavior of water molecule inside the receptor. Structures often show isolated water molecule in interaction with the protein and give no clue about the global hydration of the receptor's pockets. MD simulations of inactive and active state of GHSR allowed us to analyze a fully solvated receptor, where the global hydration pattern varied as a function of the conformational state of the receptor. Even though we simulated the wild-type GHSR, sidechain positions and hydration patterns in both states were compatible with our experiment for positions $134^{3.43}$, $232^{5.58}$, $268^{6.40}$, and $272^{6.44}$. In contrast, however, the hydration at position $315^{7.45}$ did not agree with our experimental results. A possibility would be that the size of the residue itself excludes water molecules from its vicinity, and/or orients it toward the membrane, thus explaining the low hydration in the inactive and the active states of the receptor we measured experimentally.

Interestingly, a difference in the GHSR hydration pattern was observed depending on whether the ligand was a full or a Gq-biased agonist. This indicates a different arrangement of the water network in the conformational states stabilized by these ligands. This is in line with previous observations with NK1R where mutation of the water hydrogen bond network affected Gq- and Gs-mediated signaling in a different way (*Valentin-Hansen et al., 2015*). In the same way, the central polar network in the GLP1 receptor has been suggested to be critical for G protein-dependent but not for G protein-independent signaling (*Wootten et al., 2016a*). Taken together, this data indicates that the different states in the GHSR conformational landscape differ in their hydration pattern, as stated above but, in addition, that ligands with different pharmacological profiles, here a full and a partial, biased agonist, have a different impact on the distribution of these states. This conclusion with GHSR is consistent with our previous data using a different conformational reporter, monobromobimane (*Mary et al., 2012*).

In addition to ligands, other components in the receptor environment allosterically impact on GPCR activation and conformational dynamics. This is the case of the lipid bilayer whose composition has been shown to impact on the structure and function of many different membrane proteins including ion channels (*Hille et al., 2015*) and receptors (*Strohman et al., 2019*; *Dawaliby et al., 2016*; *Casiraghi et al., 2016*). Among the lipids reported to affect GPCR signaling such as cholesterol (*Casiraghi et al., 2016*; *Zocher et al., 2012*) or charged phospholipids (*Strohman et al., 2019*; *Dawaliby et al., 2016*), PIP2 has been shown to stabilize the receptor:G protein complex for the adenosine A2a-, $\beta_1$-adrenergic, and neurotensin receptor 1 (*Yen et al., 2018*). More recently, we showed that PIP2 could be an allosteric regulator of ghrelin signaling (*Damian et al., 2021*). Accordingly, we observed here that including PIP2 in the nanodiscs was associated with a further amplification of the changes in 7H4MC fluorescence triggered by ghrelin, reflecting the allosteric coupling between the full agonist and this lipid for stabilizing an active-like hydration pattern of GHSR.

In closing, the combination of incorporation of 7H4MC-ethylglycine into proteins, fluorescence spectroscopy, and advanced MD simulations provided us with a straightforward strategy to delineate conformational events associated with GHSR activation through an unexplored but nevertheless central feature in the functioning of membrane proteins, local hydration. Using this strategy, we found that the hydration pattern in specific regions of TM5 and TM6 is dependent on the activation

state of the receptor. This illuminates an unexpected role of water molecules as possible allosteric modulators of GHSR activation, consistent with their general effect on the allosteric regulation of proteins (*Leitner et al., 2020*). Hence, a model emerges where the activation process of GPCRs and their final signaling output could be the result of the concerted, synergistic, and exquisitely tuned influence of all the components in the receptor environment, including the solvent, on the distribution of the different states composing their conformational landscape. In addition, these observations demonstrate that water movements are tightly correlated to the receptor activation process and could therefore be used as a fingerprint to navigate the conformational landscape of GPCRs.

# Materials and methods

## Key resources table

| Reagent type (species) or resource | Designation | Source or reference | Identifiers | Additional information |
|---|---|---|---|---|
| Strain, strain background | BL21(DE3) *Escherichia coli* | Sigma-Aldrich | CMC0014 | Chemically competent cells |
| Recombinant DNA reagent | pEvol-aaRS | doi: 10.1021/ja062666k | | |
| Recombinant DNA reagent | pMSP1E3D1 | Addgene | #20066 | |
| Recombinant DNA reagent | pET21a-α5-GHSR (transfected construct; *Homo sapiens*) | doi: 10.1074/jbc.M111.288324 | | |
| Peptide, recombinant protein | Ghrelin | This work | | Synthesis is described in the Materials and methods section |
| Peptide, recombinant protein | Fluorescent ghrelin | This work | | Labeling is described in the Materials and methods section |
| Peptide, recombinant protein | Thrombin | Sigma | T7009 | |
| Commercial assay or kit | GTPase-GloTM assay | Promega | V7681 | |
| Chemical compound, drug | 7H4MC-ethylglycine | This work | | Synthesis is described the Materials and methods section |
| Chemical compound, drug | Ampicillin | Sigma | A9518 | |
| Chemical compound, drug | Chloramphenicol | Calbiochem | 220551 | |
| Chemical compound, drug | IPTG | Sigma | I6758 | |
| Chemical compound, drug | Amphipol A8-35 | Anatrace | A835 100 MG | |
| Chemical compound, drug | β-DDM | Anatrace | D310 | |
| Chemical compound, drug | Cholesteryl-hemisuccinate | Anatrace | CH210 | |
| Chemical compound, drug | POPC | Avanti Polar Lipids | 850457C | |
| Chemical compound, drug | POPG | Avanti Polar Lipids | 840457C | |

*Continued on next page*

*Continued*

| Reagent type (species) or resource | Designation | Source or reference | Identifiers | Additional information |
|---|---|---|---|---|
| Chemical compound, drug | PIP2 | Avanti Polar Lipids | 850155P | |
| Chemical compound, drug | Bio-Beads SM-2 | BIO-RAD | 1528920 | |
| Chemical compound, drug | Lumi4-Tb NHS | CisBio | 62TBSPEA | |
| Chemical compound, drug | DY647P1-maleimide | Dyomics | 647P1-03 | |
| Chemical compound, drug | Amine reactive Tb chelate | Fisher | 11563467 | |
| Chemical compound, drug | NiNTA Superflow | Qiagen | 30430 | |
| Chemical compound, drug | Streptavidin-agarose | Thermofisher | 20361 | |
| Chemical compound, drug | Superdex S200 increase 10×300 GL | GE Healthcare (Cytiva) | 28990944 | |
| Chemical compound, drug | Source 15Q 4.6×100 PE | GE Healthcare (Cytiva) | 17518101 | |
| Chemical compound, drug | ZebaSpin 40K MWCO column | Thermofisher | 87766 | |
| Software, algorithm | Prism | GraphPad | Version 8.4.3 | |
| Software, algorithm | VMD | doi: 10.1016/0263-7855(96)00018-5 | | |
| Software, algorithm | Bio3D | doi: 10.1093/bioinformatics/btl461 | | |
| Software, algorithm | Pymol | Schrodinger LLC | | |
| Software, algorithm | Gromacs 2020.3 | doi: 10.5281/zenodo.3923645 | | |

## Materials

MSP1E3D1(-) was expressed and purified in *E. coli* as described (*Ritchie et al., 2009*). 7H4MC-ethylglycine was synthesized as described (*Amaro et al., 2015*) with the exception that the final product was purified using reverse-phase HPLC.

## Production of 7H4MC-labeled GHSR

For labeling with 7H4MC-ethylglycine, the TAG amber codon was introduced at the positions indicated in *Figure 1A* by site-directed mutagenesis with the pET21a expression vector encoding human GHSR fused to the α5 integrin (*Damian et al., 2012*). The UAA solution was prepared by dissolving 263 mg of 7H4MC-ethylglycine in 10 mL 200 mM KOH solution and filter-sterilizing. The ghrelin receptor expression vector was co-transformed with the pEvol-aaRS carrying the engineered orthogonal tRNA and aminoacyl-tRNA synthase pair (*Wang et al., 2006*) in BL21(DE3) *E. coli* cells. Cultures were grown at 37°C in 2YT medium containing ampicillin and chloramphenicol until the OD600 reached 0.5–0.6. After centrifugation, cell pellets were resuspended in fresh 2YT-ampicillin-chloramphenicol medium containing 10 mL of the UAA solution. The culture was incubated again at 37°C until OD600 reached 1 and protein expression was induced by addition of IPTG and arabinose (1 mM and 0.02%, respectively). Cell growth was continued for 16 hr at 30°C. In all cases, GHSR purification and assembly into nanodiscs was carried out as described for the unlabeled receptor (*Damian et al., 2012*). Briefly, the α5-GHSR fusion protein was first purified from inclusion bodies as an SDS-unfolded protein using IMAC. After cleavage of the α5 integrin partner with thrombin, the

resulting receptor was dialyzed in a 50 mM Tris-HCl, 1% SDS, pH 8 buffer. Amphipol (APol)-mediated folding was then carried out by adding APol A8-35 to the SDS-solubilized receptor at a 1:5 protein/APol weight ratio in the presence of 10 µM of JMV3011. After 30 min incubation at room temperature, GHSR folding was initiated by precipitating dodecyl sulfate as its potassium salt through addition of KCl to a final 200 mM concentration. The potassium dodecyl sulfate precipitate was then removed by two 15 min centrifugations at 16,100×g. The supernatant was extensively dialyzed against a 50 mM potassium phosphate, 150 mM KCl, 10 µM JMV3011, pH 8 buffer. APols were then exchanged to n-dodecyl-β-D-maltopyranoside (β-DDM) in the presence of cholesteryl hemisuccinate (CHS). To this end, the APol/GHSR complex was incubated for 2 hr at 4°C with 0.2% (w/v) β-DDM, 0.02% (w/v) CHS in a 50 mM Tris-HCl pH 8, 150 mM NaCl, 10 µM of the JMV3011 buffer. The sample was then loaded onto a pre-equilibrated HisTrap column and the resin washed with a 50 mM Tris-HCl pH 8, 150 mM NaCl, 0.2% (w/v) β-DDM, 0.02% (w/v) CHS, 10 µM JMV3011 buffer and then with a 50 mM Tris-HCl pH 8, 150 mM NaCl, 0.1% (w/v) β-DDM, 0.02% (w/v) CHS, 10 µM JMV3011 buffer. The protein was finally eluted from the column with the same buffer containing 200 mM imidazole and dialyzed into a 25 mM HEPES, 100 mM NaCl, 2 mM β-DDM, 0.02% (w/v) CHS, 10 µM JMV3011 buffer. For reconstitution into nanodiscs, the His-tagged receptor in 25 mM HEPES, 100 mM NaCl, 2 mM β-DDM was first bound onto a pre-equilibrated Ni-NTA superflow resin at a protein-to-resin ratio at 0.1–0.2 mg of receptor per mL of slurry (batch conditions). The receptor was then mixed with 10 µM of JMV3011, and with MSP1E3D1(-) and a POPC:POPG (3:2 molar ratio) mixture, in the absence or presence of PIP2 (2.5% PIP2-to-total lipid molar ratio), at a 0.1:1:75 receptor:MSP:lipid ratio, with the receptor still immobilized on the Ni-NTA matrix. After 1 hr incubation at 4°C, polystyrene beads (Bio-Beads SM-2) were added to the slurry at an 80% (w/v) ratio and incubated under smooth stirring for 4 hr at 4°C. The resin was then extensively washed with a 50 mM Tris-HCl pH 8, 150 mM NaCl buffer, and the His-tagged receptor eluted with the same buffer containing 200 mM imidazole. After extensive dialysis in a 25 mM HEPES, 150 mM NaCl, 0.5 mM EDTA, pH 7.5 buffer, active receptor fractions were purified using affinity chromatography (*Ferré et al., 2019*). To this end, the receptor in lipid discs was loaded on a streptavidin-agarose column where the biotinylated JMV2959 antagonist had been bound following manufacturer's instructions. After washing with 25 mM Tris-HCl, 150 mM NaCl, pH 7.4, the bound proteins were recovered by washing the column with the same buffer containing 1 mM of the low affinity JMV4183 antagonist. This antagonist was then removed through extensive dialysis against a 25 mM Tris-HCl, 150 mM NaCl, 0.5 mM EDTA, pH 7.4 buffer. We previously demonstrated that under such conditions all the ligand is removed from its binding site on GHSR (*Ferré et al., 2019*). Homogeneous fractions of GHSR-containing discs were finally obtained through a size-exclusion chromatography step on an S200 increase column (10/300 GL) using the 25 mM Tris-HCl, 150 mM NaCl, 0.5 mM EDTA, pH 7.4 buffer as the eluent (*Figure 1—figure supplement 1*).

## Receptor labeling for ligand-binding assays

To avoid any labeling of the scaffolding protein, labeling of the receptor N-terminus with the amine-reactive Tb chelate for the HTRF-monitored ligand-binding assays was carried out in the APol-folded state (*Damian et al., 2012*), that is, before insertion into the nanodiscs. To this end, the receptor in A8-35 was dialyzed in a 50 mM potassium phosphate, 100 M KCl, pH 7.7 buffer. This pH value was determined from a series of labeling reactions we first carried out at different pH to define the optimal value for labeling essentially the protein N-terminal α-amine and not the lysyl ε-amino groups (*Damian et al., 2012*), which display a higher pKa value (*Grimsley et al., 2009*). The amine-reactive chelate was added to the protein solution (dye-to-protein equimolar ratio), and the reaction was carried out overnight at 4°C under constant stirring. The conjugate was separated from any possible unreacted labeling reagent by desalting on a ZebaSpin 40K column. Specific labeling of the N-terminal amine was assessed in a pilot experiment by the absence of fluorescence of the labeled receptor after digestion with TEV of a construct we designed to determine if labeling indeed occurred essentially at the GHSR N-terminus (*Figure 1—figure supplement 4*). The receptor reconstitution procedure was then continued by exchanging the APol to β-DDM and assembly into nanodiscs, as described above.

## Ghrelin1-18-DY647P1 synthesis

The structure of the fluorescent ghrelin peptide we used in the ligand-binding experiments is shown in *Figure 1—figure supplement 5*. H$^1$Gly-$^2$Ser-$^3$Asp(n-octanoyl)-$^4$Phe-$^5$Leu-$^6$Ser-$^7$Pro-$^8$Glu-$^9$His-$^{10}$Gln-$^{11}$Arg-$^{12}$Val$^{13}$Gln-$^{14}$Gln-$^{15}$Arg-$^{16}$Lys-$^{17}$Glu-$^{18}$Ser-$^{19}$Cys-NH$_2$ was synthesized by solid-phase peptide synthesis starting from Agilent Amphisphere 40 RAM resin using Fmoc chemistry, HATU/DIEA system for coupling, and piperidine/DMF for deprotection. All coupling steps (5 eq.) were performed twice for 10 min, except for $^{15}$Gln, $^{12}$Val, and $^{11}$Arg where the first coupling lasted 45 min. Final deprotection was performed with a TFA/TIS/H$_2$O (95/2.5/2.5) mixture for 3 hr. After purification by preparative RP-HPLC, the peptide (0.845 eq.) was dissolved in 1 mL of sodium phosphate solution (pH 7) and 1 mL of acetonitrile and conjugated with 1 mg of DY-647P1-maleimide (Dyomics) for 3 hr. The fluorescent peptide was directly injected on a preparative RP-HPLC column and purified (*Figure 1—figure supplement 5*). Their identity and purity were evaluated by mass spectrometry analyses (*Figure 1—figure supplement 5*). Preparative RP-HPLC was run on a Gilson PLC 2250 Purification system instrument (Villiers le Bel, France) using a preparative column (Waters DeltaPak C18 Radial-Pak Cartridge, 100 Å, 40–100 mm, 15 µm particle size) in gradient mode with a flow rate 50.0 mL/min. Buffer A was 0.1% TFA in water, and buffer B was 0.1% TFA in acetonitrile.

## LC/MS analyses

The LC/MS system consisted of an HPLC-ZQ (Waters) equipped with an ESI source. Analyses were carried out using a Phenomenex Kinetex column (C18, 100 Å, 100×2.1 mm$^2$, 2.6 µm). A flow rate of 0.5 mL/min and a gradient of 0–100% B in 5 min were used: eluent A, water/0.1% HCO$_2$H; eluent B, ACN/0.1% HCO$_2$H. Positive ion electrospray (ESI+) mass spectra were acquired from 100 to 1500 m/z with a scan time of 0.2 s. Nitrogen was used for both the nebulizing and drying gas.

## MALDI MS and MS/MS analyses

Samples were analyzed from CHCA or SA matrix deposits, in positive ion mode with a Rapiflex (Bruker Daltonics) instrument. A pulsed Nd:YAG laser at a wavelength of 355 nm was operated at a 66.7 Hz frequency with a laser focus of 29%. Data were acquired with the Flex Control software (version 4.1, Bruker Daltonics). Spectra were integrated with the Flex Analysis software (version 4.0, Bruker Daltonics), the centroid algorithm was used to assign peaks. An acceleration voltage of 25.0 kV (IS1) was applied for a final acceleration of 21.95 kV (IS2) and lense voltage of 9.6 kV. The reflectron mode was used for the ToF analyzer (voltages of 26.3 and 13.8 kV). The delayed extraction time was 30 ns. Acquisitions were performed using a reflector detector voltage of 1.722 kV. MS data were processed with the Flex Analysis software (version 4.0, Bruker Daltonics). External calibration was performed with commercial peptide mixture (Peptide Calibration Standard II, Bruker Daltonics). Fragmentation experiments were performed under laser-induced dissociation conditions with the LIFT cell voltage parameters set at 19.0 kV (LIFT 1) and 3.7 kV (LIFT 2) for a final acceleration of 29.5 kV (reflector voltage) and a pressure in the LIFT cell around $4 \times 10^{-7}$ mbar. The precursor ion selector was set manually to the first monoisotopic peak of the molecular ion pattern for all analyses. MS/MS data were processed with the Flex Analysis software (version 4.0, Bruker Daltonics). Mass lists were generated according to the following parameters: SNAP as peak detection algorithm, S/N threshold 3.

## G protein production

A G$\alpha_q\beta_1\gamma_2$ heterotrimer composed of the wild-type rat G$\alpha_q$ and bovine G$\beta_1$ subunits and of a bovine G$\gamma_2$ subunit tagged with a hexahistidine was expressed in *sf9* cells and purified as described (*Kozasa, 2004*). For the functional assay, the protein was further purified by ion-exchange chromatography. To this end, the heterotrimer was isolated using a Source 15Q 4.6×100 PE column. After binding of the protein to the column in a 20 mM HEPES, 30 mM sodium chloride, 1 mM MgCl$_2$, 0.05% DDM, 100 mM TCEP, 20 mM GDP, pH 7.5 buffer and washing with the same buffer, the heterotrimer was eluted with a linear gradient of 30–500 mM NaCl and the fractions containing the G protein trimer were pooled (see SDS-PAGE profile in *Figure 1—figure supplement 6*).

## Functional assays

Competition ligand-binding assays were performed using fluorescence energy transfer with a purified receptor labeled at its N-terminus with Lumi-4 Tb NHS and the dy647-labeled ghrelin peptide (*Damian et al., 2015*; *Leyris et al., 2011*). Increasing concentrations in the competing compound were added to a receptor:ghrelin peptide mixture (100 nM concentration range). After a 30 min incubation at 15°C, fluorescence emission spectra were recorded at the same temperature between 500 and 750 nm (Cary Eclipse spectrofluorimeter, Varian) with excitation at 337 nm. GTP turnover was assessed as described (*Hilger et al., 2020*). All experiments were carried out at 15°C. The receptor (200 nM) was first incubated with the isolated G protein (500 nM) and, when applicable, the ligand (10 µM) for 30 min in a 25 mM HEPES, 100 mM NaCl, 5 mM MgCl$_2$, pH 7.5 buffer. GTP turnover was then started by adding GTP (1 µM) and the remaining amount was assessed after 15 min incubation at 15°C using the GTP-Glo assay (Promega).

## 7-H4MC fluorescence measurements

Fluorescence spectra were recorded with a Cary Eclipse spectrofluorimeter (Varian) equipped with a Peltier-based temperature control device. All experiments were carried out at 15°C. The emission spectra after excitation at 320 nm were recorded between 340 and 600 nm. The normalized emission intensity was fitted by means of nonlinear least-square procedure to the sum of peak function (*Amaro et al., 2015*). The R-square parameter was used to estimate the goodness of the fit.

## Statistical analyses

Data in different conditions were compared by one-way ANOVA followed by Dunnett's multiple comparison test and reporting of multiplicity-adjusted p-values and confidence intervals (*Michel et al., 2020*). As stated in the legends of the corresponding figures, data are presented as mean ± SD of three experiments. All analysis steps, including the sample size, were decided before looking at the data. No data was removed from the analysis. No measure to avoid experimental bias was taken.

## Modeling

### Building of an active-like model of GHSR

The structure of GHSR was first retrieved from the PDB (6KO5) (*Shiimura et al., 2020*) and used as a starting point for our study. We mutated back to wild-type amino acids the two mutations (T130$^{3.39}$K and N188Q) that were present in the structure to match the wild-type sequence, and modeled the extracellular loop three ab initio (ECL3 – residue G293 to I300), with MODELLER 9.19 (*Webb and Sali, 2016*). Two cryo-EM structures of the ghrelin receptor in complex with ghrelin or a synthetic agonist and a Gq mimetic have been posted on the BioRxiv preprint server (https://doi.org/10.1101/2021.06.09.447478). However, the coordinates of these models are not yet available. Hence, in order to capture differences in the receptor hydration pattern upon activation, we generated an active-like model of GHSR by targeted molecular dynamics (TMD) simulations performed in an explicit membrane environment. First, we modeled the target conformation based on the dopamine D2 receptor coupled to Gi (*Yin et al., 2020*) (D2R:Gi, PDB id: 6VMS; sequence similarity: 33%) by homology modeling using MODELLER 9.19 (*Webb and Sali, 2016*). The sequence alignment between GHSR and D2 was achieved with ClustalW (*Larkin et al., 2007*). The best out of 100 models built by MODELLER, regarding DOPE score, was further selected as the target conformation for the subsequent TMD. The TMD simulation was run with NAMD 2.13 (*Phillips et al., 2020*), where the inactive experimental conformation was pushed toward the newly generated active conformation. Inactive GHSR was embedded in a lipid bilayer containing 156 POPC (1-palmitoyl-2-oleoyl-sn-glycero-3-phosphocholine), for a size of 80×80 Å$^2$. The system was then solvated and neutralized with a NaCl concentration around 0.15 M (17,270 water molecules, 46 sodium, and 29 chloride ions) with CHARMM-GUI (*Wu et al., 2014*; *Brooks et al., 2009*; *Jo et al., 2008*). In order to limit the deviation from the initial experimental structure, the force during TMD was only applied to residues of the intracellular part of TM helix 6 (TM6, from S252$^{6.24}$ to L277$^{6.49}$), which are known to undergo the largest conformational changes during activation of all known GPCRs. All remaining atoms of GHSR were harmonically restrained in position using a force constant of 1 kcal/mol/Å$^2$, but residues L239$^{5.65}$ to A251 (ICL3), so that the loop could follow the motion of TM6. Prior to TMD, the system

was minimized using 10,000 steps of conjugate gradient as implemented in NAMD 2.13 (*Phillips et al., 2020*), followed by successive short equilibration procedures in NVT and NPT ensembles, to reach a final temperature of 300 K and a pressure of 1 bar using CHARMM36m force field (*Huang et al., 2017*). We did not modify the equilibration procedure designed by CHARMM-GUI developers (*Wu et al., 2014*). The TMD simulation was performed in the NPT ensemble (300 K and 1 bar) over a period of 500 ps using a force constant of 200 kcal/mol/$\text{Å}^2$ scaled down by the number of selected atoms in TM6 (477 atoms including hydrogens). Non-bonded interactions were truncated at a distance cut-off of 12 Å applying a switching function in the range 10–12 Å, while long range electrostatics were computed via particle mesh Ewald (PME).

## MD simulations of inactive and active-like conformers of wild-type GHSR

The inactive (experimental) and active (modeled) conformers of GHSR were simulated by MD with Gromacs 2020.3 using the CHARMM36m force field (*Huang et al., 2017*). To fit to the experimental membrane composition used in this study, each conformer was embedded in a symmetric lipid bilayer of size 80×80 $\text{Å}^2$, where each layer was composed of 20 cholesterol, 28 1-palmitoyl-2-oleoyl-sn-glycero-3-phosphoglycerol (POPG), 42 POPC, and 10 phosphatidylinositol-4,5-bisphosphate (PIP2) (including five PIP2 protonated on one phosphate group and five PIP2 protonated on the other phosphate group, named respectively POPI24 and POPI25 in CHARMM36m force field). Systems were solvated and their charges were neutralized with a NaCl concentration around 0.15 M (17,751 water, 170 sodium, and 47 chloride ions). The simulation setup was done with the CHARMM-GUI webserver (*Wu et al., 2014*; *Brooks et al., 2009*; *Jo et al., 2008*). Contrary to the TMD protocol, we did change the default CHARMM-GUI procedure for equilibration. Indeed, we added three additional equilibration steps to the default CHARMM-GUI procedure. We modified the harmonic restraints on atomic positions and the number of simulation steps to allow a smooth relaxation of the systems (*Supplementary file 1*). We reproduced this protocol five times for each system (active and inactive) modifying the starting velocities so that the convergence of the resulting data could be discussed. The production was run in the NPT ensemble (300 K and 1 bar) for 5 µs (leading to a simulation time of 50 µs in total). It is important to notice that, during production, all restraints and constraints were removed. For all simulations, direct non-bonded interactions were truncated at a distance cut-off of 12 Å applying a switching function in the 10–12 Å range, while long range electrostatics were computed via PME.

## PCA of experimental structures

To delineate the possible motions described by the plethora of available GPCRs' experimental structures, we retrieved 268 structures of class A GPCRs from the PDB. To homogenize these data, only the part corresponding to a single isolated receptor was conserved for further analysis, for instance removing the intra- and/or extracellular partner(s) if required or other copies of the same receptor in the case of dimeric structures. The sequences of all retrieved structures were then aligned with Clustal Omega (*Larkin et al., 2007*) with default parameters. To perform PCA of the resulting set of coordinates, the length of the resulting sequences also required to be homogenous. As a compromise between the number of structures considered (increasing the conformational diversity) and the length of the sequence common to all receptors (improving the structural description), we decided to discard a residue at a particular position of the alignment if the latter was missing in at least two structures out of the 268. In addition, a structure was discarded if it was the only one presenting a missing residue at a specific position. Using these criteria, only six structures were deleted from the initial set (PDB id: 5WB2, 4PY0, 5ZKP, 3RZE, 4RWA, and 4DAJ). In summary, 262 structures were considered, together describing a set of 164 conserved amino acids (GHSR numbering: 45, 46, 48–68, 76–102, 120–148, 162–179, 181, 212, 213, 215–219, 221–239, 261, 263–286, 310–324). The list of the considered PDB structures together with useful information, according to GPCRdb (*Pándy-Szekeres et al., 2018*), can be found in *Supplementary file 2*. Not surprisingly, the final selection covered most of the TM domains, ensuring a good description of the internal motions coded by the ensemble of experimental structures (*Figure 4—figure supplement 5*). On the contrary, most residues located in the extra- or intracellular loops were excluded. Because of the variability of residues at each position of the final alignment, the PCA was performed only on the coordinates of the Cα atoms with the R package Bio3D (*Grant et al., 2006*).

### Analysis and figure generation

All analyses were run with VMD (*Humphrey et al., 1996*) and the R package Bio3D (*Grant et al., 2006*). Figures were generated using VMD and Pymol.

## Acknowledgements

PG Schultz and L Supekova (TSRI, La Jolla, CA) are greatly acknowledged for providing genetic material. This work was supported by CNRS, Université de Montpellier, Agence Nationale de la Recherche (ANR-17-CE11-0011, ANR-17-CE11-22, ANR-17-CE18-0022), EpiGenMed Labex (post-doctoral fellowship to KBH) and DYNAMO Labex (post-doctoral fellowship to MC). This program received funding from the European Union's Horizon 2020 research and innovation programme under the Marie Sklodowska-Curie grant agreement n° 799376. Mass spectrometry analyses were performed on the instruments located in the IBMM platform of instrumentation, Laboratoire de Mesures Physiques (LMP) of Université de Montpellier. We thank GENCI (Grand Équipement National de Calcul Intensif), CINES (Centre Informatique National de l'Enseignement Supérieur), and IDRIS (Institut du développement et des ressources en informatique scientifique) for computational resources. We also thank CAMPUS FRANCE for promoting the French-Brazilian collaboration via the CAPES-COFECUB project number Ph-C882/17.

## Additional information

### Funding

| Funder | Grant reference number | Author |
|---|---|---|
| Agence Nationale de la Recherche | ANR-17-CE11-0011 | Jean-Louis Banères |
| Agence Nationale de la Recherche | ANR-17-CE18-0022 | Jean-Alain Fehrentz |
| Labex | EpiGenMed | Khoubaib Ben haj salah |
| Campus France | Ph-C882/17 | Nicolas Floquet |
| Agence Nationale de la Recherche | ANR-17-CE11-0022 | Jean-Louis Banères |
| European Union's Horizon 2020 | Marie Sklodowska-Curie grant agreement No 799376 | Marina Casiraghi |

The funders had no role in study design, data collection and interpretation, or the decision to submit the work for publication.

### Author contributions

Maxime Louet, Conceptualization, Data curation, Formal analysis, Validation, Investigation, Writing - review and editing; Marina Casiraghi, Formal analysis, Investigation, Writing - review and editing; Marjorie Damian, Sonia Cantel, Severine Denoyelle, Data curation, Formal analysis, Investigation, Writing - review and editing; Mauricio GS Costa, Pedro Renault, Paulo R Batista, Data curation, Formal analysis, Methodology, Writing - review and editing; Antoniel AS Gomes, Sophie Mary, Data curation, Formal analysis; Céline M'Kadmi, Data curation, Formal analysis, Investigation; Khoubaib Ben Haj Salah, Investigation; David Perahia, Paulo M Bisch, Formal analysis, Methodology, Writing - review and editing; Jean-Alain Fehrentz, Formal analysis, Supervision, Writing - review and editing; Laurent J Catoire, Data curation, Formal analysis, Supervision, Writing - original draft; Nicolas Floquet, Conceptualization, Formal analysis, Supervision, Methodology, Writing - original draft; Jean-Louis Banères, Conceptualization, Formal analysis, Supervision, Funding acquisition, Writing - original draft, Project administration, Writing - review and editing

### Author ORCIDs

Marina Casiraghi (iD) https://orcid.org/0000-0003-0627-2288
Mauricio GS Costa (iD) http://orcid.org/0000-0001-5443-286X

Paulo R Batista ⒾD http://orcid.org/0000-0002-3399-174X
Khoubaib Ben Haj Salah ⒾD https://orcid.org/0000-0002-2313-2773
Jean-Alain Fehrentz ⒾD https://orcid.org/0000-0002-6064-3118
Nicolas Floquet ⒾD https://orcid.org/0000-0002-3883-2852
Jean-Louis Banères ⒾD https://orcid.org/0000-0001-7078-1285

**Decision letter and Author response**
Decision letter https://doi.org/10.7554/eLife.63201.sa1
Author response https://doi.org/10.7554/eLife.63201.sa2

## Additional files

### Supplementary files
• Supplementary file 1. Equilibration procedure.
• Supplementary file 2. List of structures used for principal component analysis.
• Transparent reporting form

### Data availability
All data generated or analysed during this study are included in the manuscript and supporting files.

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
