## [Decision Letter]

**Acceptance summary:**

This manuscript uses un-natural amino acid incorporation into the GHSR1a to examine the exposure of particular residues to changes in polarity (interpreted as solvent exposure) experimentally followed by molecular dynamics simulation. Considering the body of work done on GPCRs, there are surprisingly few studies that carry out a quantitative one-to-one comparison between experimental and simulations. This manuscript presents a convincing attempt at doing so.

**Decision letter after peer review:**

Thank you for submitting your article "Concerted conformational dynamics and water movements in the ghrelin G protein-coupled receptor" for consideration by *eLife*. Your article has been reviewed by 3 peer reviewers, including Lucie Delemotte as the Reviewing Editor and Reviewer #1, and the evaluation has been overseen by a Reviewing Editor and Richard Aldrich as the Senior Editor.

The reviewers have discussed the reviews with one another and the Reviewing Editor has drafted this decision to help you prepare a revised submission.

Summary:

This manuscript uses un-natural amino acid incorporation into the GHSR1a to examine the exposure of particular residues to changes in polarity (interpreted as solvent exposure) experimentally followed by molecular dynamics simulation.

Considering the body of work done on GPCRs, there are surprisingly few studies that attempt a quantitative one-to-one comparison between experimental and simulations. This manuscript presented an interesting attempt at doing so. However, the manuscript asks the reader to accept a substantial amount of work that has either obscure, or superficial methods, little supporting data and significant normalization, making it difficult to fully judge its merits.

Essential revisions:

Related to experiments:

1) Please address the functionality of the expressed, refolded and MSP incorporated GHSR1a. Could the authors please provide original Coomassie stained gels and size exclusion chromatography traces that support their purification and re-incorporation of GHSR1a into the MSP scaffold? The authors report that they have expressed GHSR1a in bacteria then unfolded and refolded this protein, the reviewers are not aware of any other GPCR for which this has been successfully performed, indeed the structural paper that the authors discuss (below), uses a thermostabilized GHSR1a expressed in insect cells in order to obtain crystallizable protein. Additionally, the below paper uses a thermofluor assay to demonstrate the stabilization of their construct. The thermofluor assay, by its nature, indicates that GHSR1a does not spontaneously refold.

Shiimura Y, Horita S, Hamamoto A, Asada H, Hirata K, Tanaka M, Mori K, Uemura T, Kobayashi T, Iwata S, Kojima M. Structure of an antagonist-bound ghrelin receptor reveals possible ghrelin recognition mode. Nat Commun. 2020 Aug 19;11(1):4160. doi: 10.1038/s41467-020-17554-1. PMID: 32814772; PMCID: PMC7438500.

2) As evidence of the functionality of the refolded, purified and MSP incorporated GHSR1a the authors provide a FRET competition binding assay. There is insufficient explanation of this assay for this to be reproduced by another group. In the method the authors indicate that the GHSR1a is labelled at the N-terminus with Lumi-4 Tb NHS. NHS esters typically react with any primary amine, in the authors methods they have the purified GHSR1a in a 25 mM Tris buffer, where the Tris would be expected to preferentially react with the Lumi-4 Tb NHS, the GHSR1a is also incorporated into MSP, which has an N-terminus and both proteins have a number of lysine residues where the Lumi-4 Tb NHS would be expected to react with the epsilon amino group. Perhaps there are some significant details missing from their methods that might enlighten this? In any case, could the authors please provide their in-gel fluorescence (or alternative analysis such as mass spectrometry) that demonstrates specific labelling of GHSR1a (and not MSP) on the N-terminus (and not on lysine side chains)? This assay uses a dy647 labelled ghrelin peptide, could the authors please provide details of how the labelling was performed and either HPLC of mass spec data for the resulting labelled reagent? The cited reference (19) does not appear to contain this reagent, whereas the cited reference (33) does contain a "red-ghrelin" where no details about the chemical composition are readily available. The published affinity of ghrelin for GHSR1a is 400 pM (https://www.guidetopharmacology.org/GRAC/ObjectDisplayForward?objectId=246), the authors need to specifically address why the reported affinity in their assay (Figure 1B) appears to be approximately 250 fold lower at ~100 nM? Could the authors also please provide the original, non-normalised FRET data so that the reader can understand the window for this assay along with a specificity control such as Lumi-4 Tb NHS labelled empty MSP nanosdiscs?

3) In figure 1C the authors provide further evidence for functionality of their purified GHSR1a using a GTP turnover assay. The authors need to provide the full details of how the Galpha/β/γ heterotrimer were expressed and purified for this assay. Please include details of which particular isoform and species each G protein subunit is from, the expression system and how they were purified. A representative Coomassie stained gel that demonstrates stoichiometric equivalence of the subunits in the purified complex should also be supplied. The references provided for this assay to not appear to relate to a GTP depletion style assay as appears to be described here. Could the authors also please describe how 0% was defined for the assay and the relative concentrations of GHSR1a and G proteins heterotrimer that were added to the reactions? The authors show apparent differences in bound GTP in this assay, could they please provide a statistical analysis for these differences?

Related to simulations:

1) Driving large structural changes fast is risky, and the reviewers were not convinced the water populations had equilibrated. Indeed, forces to enhance the sampling were applied along each PC separately. Since these PCs represent a linear decomposition of the overall family-wide conformational change, it didn't appear wise to enhance the sampling along them: linear decomposition of the movement could in principle result in very non-physical motions. Can the authors provide the readers with a supplementary figure showing the comparison between the starting and the 2 end structures, as well as the PDBs of the resulting structures so their quality can be checked?

Relatedly, the mix of active-like and inactive-like structures used in the PCA to derive the biasing forces is expected to have a major effect. The authors need to explicitly list (in the supplement) the structures used, and categorize them by activity state and preferably by GPCR family as well. The first principle vector probably point more or less along the path between inactive and active, but it would be nice to check this.

2) The reviewers also asked for a clearer rationalization of why the authors picked this sampling strategy as opposed to (1) building models of the ghrelin receptor in different states and simulating them or (2) enhancing the sampling using a non-linear approach. Ultimately, what do we learn from the fact that PC1 is the most compatible with the experimental data, given that the overall motion is a combination of all the PCs? It would be wise to replicate the results with a more standard MD simulation protocol to rule out artefacts from this choice of enhanced sampling methods.

3) Additionally, the methods section was unclear about several aspects:

a) it is unclear as to how many replicates were done for each mode: if it's a single replicate for each mode, no conclusion could be drawn about the hydration. To have any confidence in the result, the reviewers would want to see the simulations rerun many times (at least 10x, perhaps more depending on how variable the answer is), preferably starting from different structures within the equilibrated ensemble.

b) Which state does the original model represent? Can the comparison to the recently published structure be more thorough than simply showing a Calpha(?) RMSD (Figure S8)? Are the enhanced sampling MD carried out in presence or absence of ligand and why? Finally the method section as well as the Results section explaining the enhanced sampling protocol should be clarified such that it is not necessary to read the original paper explaining the method to understand.

c) Many key details of the simulations are missing: number of lipids, number of waters, electrostatics method (as written, it sounds like they didn't use Ewald, which would be a huge problem).

d) There is no discussion of statistical convergence. The simulations are very short by today's standards, and the reviewers saw no reason to assume the protein has stopped systematically changing after 300-350 ns (given the uncertainty of starting from a homology model), let alone begun actually sampling. The only evidence is Supp Figure 5, which shows the RMSD is still increasing, while nothing at all is shown for the mutants.

e) As it stands, too many details are missing for these calculations to be repeated. Please collect and document all of the scripts used (building the system, running, and analyzing) and put them either into the supplementary info or better yet into a separate repository (e.g. GitHub, zenodo, etc).

---

## [Author Response]

Essential revisions:Related to experiments:1) Please address the functionality of the expressed, refolded and MSP incorporated GHSR1a. Could the authors please provide original Coomassie stained gels and size exclusion chromatography traces that support their purification and re-incorporation of GHSR1a into the MSP scaffold?

As requested, the size-exclusion chromatography (SEC) profile of the GHSR-containing nanodiscs, along with the SDS-PAGE profile of the fractions pooled, is presented as Figure 1—figure supplement 1 in the revised version of the manuscript. As evidenced by these SEC and SDS-PAGE profiles, the GHSR-containing discs correspond to a homogeneous fraction composed of the scaffolding protein and receptor only. To be noted, faint bands corresponding to multimeric receptor species are present in the SDS gel, but this does not reflect the occurrence of such species in the nanodiscs. Indeed, we previously demonstrated unambiguously using native gel electrophoresis and fluorescence transfer that the conditions we use for nanodisc assembly (large MSP-to-receptor molar ratio, on-matrix batch assembly with a large resin-to-protein ratio) provide lipid discs with a single receptor (1). Similar conditions were also shown to provide monomeric receptor for a series of other GPCRs, including rhodopsin and the b_2_-adrenergic receptor (2,3). Oligomerization of a membrane protein on an SDS-PAGE, as in the case of the gel in Figure 1—figure supplement 1, does not reflect its oligomerization state in solution when associated to various surfactants or lipids. Indeed, at the concentration used to run a gel, SDS is known to only partially unfold these membrane proteins and trigger some aggregation, with the exception of β-barrel ones. The consequences most frequently encountered are an anomalously electrophoretic mobility and protein aggregation in the gel (R.B. Gennis “Biomembranes: Molecular Structure and Function”, Springer Science, 1989, page 93"; Rath et al. (2009) (4)).

The authors report that they have expressed GHSR1a in bacteria then unfolded and refolded this protein, the reviewers are not aware of any other GPCR for which this has been successfully performed, indeed the structural paper that the authors discuss (below), uses a thermostabilized GHSR1a expressed in insect cells in order to obtain crystallizable protein. Additionally, the below paper uses a thermofluor assay to demonstrate the stabilization of their construct. The thermofluor assay, by its nature, indicates that GHSR1a does not spontaneously refold.Shiimura Y, Horita S, Hamamoto A, Asada H, Hirata K, Tanaka M, Mori K, Uemura T, Kobayashi T, Iwata S, Kojima M. Structure of an antagonist-bound ghrelin receptor reveals possible ghrelin recognition mode. Nat Commun. 2020 Aug 19;11(1):4160. doi: 10.1038/s41467-020-17554-1. PMID: 32814772; PMCID: PMC7438500.

We agree that our production procedure, which is based on the expression of the ghrelin receptor as an unfolded protein in *E. coli* inclusion bodies followed by in vitro folding and purification of the active receptor using ligand-affinity chromatography, is not the most common one in the field. It has nevertheless been successfully used in several occasions by us and others, first for an olfactory receptor (5), and then for different GPCRs such as the receptors for neuropeptides Y1 and Y2 (6,7), ghrelin (1,8,9), chemokine (10,11), leukotriene B4 (12,13) or serotonin (14). In all cases, the receptor obtained was representative of the native one in terms of ligand binding and activation of G proteins. This is the case for the ghrelin receptor we used in the present work that, since our *princeps* paper in 2012 (1), has been repeatedly shown by us and others to be structurally and pharmacologically relevant (8,9,15-19) (see below). Besides, in our opinion, GHSR produced in *sf9* cell for crystallization purposes can be hardly compared to the present situation, as the former study dealt with a significantly different version of GHSR where the receptor was fused to BRIL at its N-terminus, devoid of its N- and C-terminal regions, thermostabilized through specific mutations and purified in a detergent solution.

2) As evidence of the functionality of the refolded, purified and MSP incorporated GHSR1a the authors provide a FRET competition binding assay. There is insufficient explanation of this assay for this to be reproduced by another group. In the method the authors indicate that the GHSR1a is labelled at the N-terminus with Lumi-4 Tb NHS. NHS esters typically react with any primary amine, in the authors methods they have the purified GHSR1a in a 25 mM Tris buffer, where the Tris would be expected to preferentially react with the Lumi-4 Tb NHS, the GHSR1a is also incorporated into MSP, which has an N-terminus and both proteins have a number of lysine residues where the Lumi-4 Tb NHS would be expected to react with the epsilon amino group. Perhaps there are some significant details missing from their methods that might enlighten this?

The description of the labeling procedure was not detailed enough, and we apologize for that. In fact, labeling of the scaffolding protein is not an issue, as the receptor is labeled before its assembly into nanodiscs, *i.e.,* in the amphipol-folded state. This procedure was described in our initial paper reporting production of GHSR in nanodiscs, although for a different version of the fluorophore (amine-reactive derivative of Alexa Fluor 350 instead of Tb chelate) (1). Before its assembly into nanodiscs, the receptor is first folded from an SDS- to an amphipol (APol)-stabilized state using a procedure we developed in close collaboration with J.-L. Popot’s laboratory (20). This procedure, which consists in exchanging SDS for APol through precipitation of the detergent as its potassium salt, allows recovery of the receptor as a stable APol:protein complex (see the SEC profile in the Figure 1—figure supplement 3B,C of the revised manuscript). Labeling with the fluorophore is then carried out at this stage. Briefly, the receptor in A8-35 APol is dialyzed in a 50 mM potassium phosphate, 100 M KCl, pH 7.7 buffer to remove any Tris salt that would indeed bias labeling, as appropriately noted by the reviewer. This pH value was determined from a series of labeling reactions we first carried out at different pH to define the optimal value for labeling essentially the protein N terminal a-amine and not the lysyl e-amino groups (1), which display a significantly higher pKa value (the average pKa values in proteins is 7.7 and 10.5 for the N-terminal a- and lysine e-amino groups, respectively (21)). The conjugate is then incubated with the Apol:protein complex under the conditions described in the Methods section and unreacted labeling reagent is removed by desalting on a ZebaSpin column (ThermoFisher). The receptor reconstitution procedure in continued by exchanging the amphipol to b-DDM and then assembling the labeled receptor into the nanodiscs. The full labeling protocol is now detailed in the revised version of the manuscript.

In any case, could the authors please provide their in-gel fluorescence (or alternative analysis such as mass spectrometry) that demonstrates specific labelling of GHSR1a (and not MSP) on the N-terminus (and not on lysine side chains)?

In-gel fluorescence may not be totally effective in assessing N-terminal specific labeling of the receptor, as we think it cannot discriminate between different positions for the fluorophore. Besides, high-resolution mass spectrometry analysis of integral a-helical membrane proteins in lipid bilayers is still not technically trivial, at least in our hands. Therefore, we applied an alternative strategy we devised previously to ascertain N-terminal labeling of G proteins with a Tb chelate (17). The experiment consists in introducing a TEV cleavage site after the protein N-terminus (see schematic representation in the figure below). This construct was used for the present experiment only. Not to affect labeling, the TEV cleavage site was introduced 11 residues after the N-terminus of the receptor. We then carried out labeling as described above and subsequently cleaved the GHSR N-terminus using TEV protease. Removal of the N-terminal residues is not detrimental to the receptor’s fold, as indicated by the crystal structure of GHSR where 28 residues were removed from the receptor N-terminus (22). The TEV cleavage is almost quantitative, as shown by the SEC profile in Figure 1—figure supplement 3. Besides, this profile shows that most of the labeling occurs at the receptor N-terminus, as the Tb-chelate moiety absorbing at 337 nm is essentially found in the column total volume after TEV digestion. This is confirmed by the UV absorption and fluorescence spectra of the non-cleaved and cleaved proteins. Indeed, essentially all the absorption and emission signatures of the Tb-chelate are lost after cleavage, indicating that most of the labeling indeed occurred at the receptor N-terminus. This figure has been added to the revised version of the manuscript as Figure 1—figure supplement 3. In any case, it is to be noted that even a residual labeling of the receptor at other sites besides the N-terminus would not be deleterious for the binding assay. Indeed, the latter simply consists in assessing the proximity between a fluorescence donor and acceptor on the receptor and the ligand, respectively, as a signature of the binding process, with no more detailed interpretation of the fluorescence transfer signal. Accordingly, equivalent binding plots have been reported using the same procedure and a fluorophore attached to any of six different positions in GHSR (23).

This assay uses a dy647 labelled ghrelin peptide, could the authors please provide details of how the labelling was performed and either HPLC of mass spec data for the resulting labelled reagent? The cited reference (19) does not appear to contain this reagent, whereas the cited reference (33) does contain a "red-ghrelin" where no details about the chemical composition are readily available.

The synthesis of the labeled peptide used in the ligand-binding assays is now described in Materials and Methods section of the revised version of the manuscript. The HPLC profile and mass spectra of the resulting product are given as Figure 1—figure supplement 4.

The published affinity of ghrelin for GHSR1a is 400 pM (https://www.guidetopharmacology.org/GRAC/ObjectDisplayForward?objectId=246), the authors need to specifically address why the reported affinity in their assay (Figure 1B) appears to be approximately 250 fold lower at ~100 nM?

Different values have been reported in the literature for the affinity of ghrelin for GHSR. It has been proposed that a peculiar behavior of ghrelin in binding assays results, at least in part, from large non-specific binding effects due to its high hydrophobicity that can lead it to bind to low-affinity ‘‘pseudo-specific” sites into the plasma membrane (24). A value as low as 400 pM has been indeed initially reported and is referenced as such in the database mentioned by the reviewer (25). As reported in the original paper, this value was obtained using radiolabeled ghrelin and human tissues, with as much as 60-65% of specific binding. However, higher values have also been reported repeatedly. For instance, a 2.3 nM Kd was obtained using ^125^I-ghrelin and GHSR-expressing COS cells (26). We ourselves measured a Kd value of 4.4 nM for ghrelin binding to GHSR using ^125^I-labeled ghrelin and the recombinant receptor expressed in HEK293 cells (24), which is close to the value reported in Holst *et al.* (2005) (26). In the same paper, we reported a Ki of 4.6 nM for ghrelin, as measured from a FRET-based assay with the receptor expressed at the surface of HEK293 cells and a fluorescent ghrelin peptide. Again, a 3-fold higher value has been recently reported for the Kd of a similarly labeled ghrelin and GHSR-expressing HEK cells (12.47 nM) (27). In any case, as perfectly noted by the reviewer, the Ki we report for the isolated receptor (77 nM, as inferred for the wild type receptor from the plot in Figure 1B) is about 15-fold lower than that measured for the same receptor in a native environment, *i.e.* in HEK293 cells (24). However, it must be kept in mind that, as is the case for most GPCRs, isolated GHSR in the absence of G proteins is in a low affinity state for agonists. Accordingly, we previously demonstrated that high affinity ghrelin binding, very similar to that observed in HEK cells, could be recovered upon reconstituting GHSR in nanodiscs with its cognate Gq protein (see Figure 1D in (19)). This data established full functionality for the ghrelin receptor assembled into nanodiscs using exactly the same procedure as the one we used in the present study.

As requested by the reviewer, the normalized binding plot has been replaced in Figure 1B by the non-normalized FRET data. As the MSP protein is not labeled with the fluorophore, a similar control experiment with empty nanodiscs could not be carried out, however. Instead, we carried out a control experiment with an unrelated receptor, the leukotriene B4 receptor, we inserted into the same nanodiscs (N-terminal labeling of the leukotriene B4 receptor with a fluorophore was initially described in (28)). This receptor is a typical class A GPCR whose general architecture is close to that of GHSR. As shown in Figure 1—figure supplement 2 of the revised manuscript, essentially no FRET was observed between BLT1-containing nanodiscs and fluorescent ghrelin, indicating very limited non-specific binding or, alternatively, that any possible residual binding of the fluorescent peptide does not contribute to the FRET signal we use to monitor receptor:ligand interaction. This plot is provided as Figure 1—figure supplement 2 in the revised version of the manuscript.

3) In figure 1C the authors provide further evidence for functionality of their purified GHSR1a using a GTP turnover assay. The authors need to provide the full details of how the Galpha/β/γ heterotrimer were expressed and purified for this assay. Please include details of which particular isoform and species each G protein subunit is from, the expression system and how they were purified. A representative Coomassie stained gel that demonstrates stoichiometric equivalence of the subunits in the purified complex should also be supplied. The references provided for this assay to not appear to relate to a GTP depletion style assay as appears to be described here.

In these functional assays, we used the heterotrimer composed of rat Ga_q_ and bovine Gb_1_g_2_. This trimer has been repeatedly used in the literature for both functional and structural studies (e.g. (29)). It was expressed in *sf9* cells using the viruses initially provided to us by T. Kozasa and T. Kawano. This system allows production of a trimer composed of Ga_q_ and Gb_1_ with a Gg_2_ subunit tagged with an hexahistidine, so that the complex can be readily purified using Ni-NTA chromatography. Purification was carried out as described in Kozasa (2004) (30). This reference has been added to the revised version of the manuscript with a description of the additional ion exchange chromatography step we used after IMAC to further purify the recombinant trimer for the functional assays. We have also included a representative Coomassie blue-stained gel as Figure 1—figure supplement 5.

Could the authors also please describe how 0% was defined for the assay and the relative concentrations of GHSR1a and G proteins heterotrimer that were added to the reactions?

Luminescence signals were normalized relative to the signal obtained under the same conditions for the Ga_q_b_1_g_2_ alone, in the absence of any receptor. A 200 nM and 500 nM concentration in receptor and G protein were used in the assays. All these data have been added in the Materials and methods section of the revised version of the manuscript.

The authors show apparent differences in bound GTP in this assay, could they please provide a statistical analysis for these differences?

A statistical analysis has been carried out and is now reported in Figure 1C of the revised version of the manuscript. This analysis shows that the difference in GTP binding in the absence and presence of agonist/inverse agonist is indeed significative. For the sake of clarity, we show in the revised version only the data for the apo, agonist and inverse agonist-loaded states, not to overload the figure. The initial figure with all the ligands (antagonist and Gq-biased agonist in addition to the full- and inverse-agonist) is now given in Figure 1—figure supplement 2.

Related to simulations:1) Driving large structural changes fast is risky, and the reviewers were not convinced the water populations had equilibrated. Indeed, forces to enhance the sampling were applied along each PC separately. Since these PCs represent a linear decomposition of the overall family-wide conformational change, it didn't appear wise to enhance the sampling along them: linear decomposition of the movement could in principle result in very non-physical motions. Can the authors should provide the readers with a supplementary figure showing the comparison between the starting and the 2 end structures, as well as the PDBs of the resulting structures so their quality can be checked?Relatedly, the mix of active-like and inactive-like structures used in the PCA to derive the biasing forces is expected to have a major effect. The authors need to explicitly list (in the supplement) the structures used, and categorize them by activity state and preferably by GPCR family as well.

We fully agree with the reviewer that functional motions are usually a combination of multiple PC. The objective here was to identify blindly all separate motions that could explain the experimental observations. It is very important to note that pulling was done with membrane and water. The pulling was slow and soft enough not to generate any nonphysical models thanks to a realistic environment (including membrane) described by a physical atomic force field (i.e. CHARMM36m). Our previous experiences with such kind of exploration along biasing vectors (e.g. normal modes) revealed that no nonphysical conformations are obtained and the structural quality of the conformational states sampled was preserved. Obtaining nonphysical motions would be related to the introduction of very strong forces, stronger than forces which maintain the topology of the receptor. In our case, we observe a wide range of amplitudes for each mode explored. This reveals that the resulting displacement does not necessarily follow precisely the biasing vector, therefore being able to diverge from the pulling direction was due to the restrictions imposed by the realistic environment. The differences in the amplitudes obtained are due to the distinct energy barriers observed for each displacement. Unfortunately, considering this very soft pulling and the statistical nature of results from MD, we were unable to reproduce our observations among different replicas using exactly the same pulling direction. We then chose to use a more classical approach by running free MD simulations in different activation states, as suggested by the reviewer.

Despite we did not use anymore the PCs inferred from experimental structures to explore GHSR motions, we used them to analyze our new MD simulations. We provide the structure list in this regard. The list of all experimental structures used to compute the PCs, together with information some readers might find useful is now available as Supplementary file 2.

The first principle vector probably point more or less along the path between inactive and active, but it would be nice to check this.

The reviewer guessed correctly the nature of the first eigenvector. It indeed represented a transition between inactive and active states and more specifically a transition between inactive adenosine receptor 2 (AA2AR) and opsin (OPSD) coupled to a G-protein (see Author response image 1).

**Author response image 1. sa2fig1:** Projections along the first-two eigenvectors inferred from a Principal Component Analysis (PCA) of all experimental structures. Names correspond to uniprot names of GPCR and have been placed in respect to the projection. Red names correspond to activated receptors, purple to intermediate conformations and blue names to inactivated receptors (according to the classification of the GPCRdb).

2) The reviewers also asked for a clearer rationalization of why the authors picked this sampling strategy as opposed to 1) building models of the ghrelin receptor in different states and simulating them or 2) enhancing the sampling using a non-linear approach. Ultimately, what do we learn from the fact that PC1 is the most compatible with the experimental data, given that the overall motion is a combination of all the PCs? It would be wise to replicate the results with a more standard MD simulation protocol to rule out artefacts from this choice of enhanced sampling methods.3) Additionally, the methods section was unclear about several aspects:a) it is unclear as to how many replicates were done for each mode: if it's a single replicate for each mode, no conclusion could be drawn about the hydration. To have any confidence in the result, the reviewers would want to see the simulations rerun many times (at least 10x, perhaps more depending on how variable the answer is), preferably starting from different structures within the equilibrated ensemble.

We initially ran 1 replica per mode, per direction and per mutant. After running two extra replicas, we could not reproduce our observations among the different replicas using the same pulling direction. This informed about how the protein could freely adapt to our soft pulling protocol and join our reply to the reviewer’s previous comment. Considering this behavior, we suspected that we would need many more than 10 replicas per pulling to observe significant trends for GHSR hydration in respect to its conformation. We thus decided to follow the reviewer’s suggestions by using a more classical approach with non-biased Molecular Dynamics simulations. We then ran 5 replicas of MD simulations (different starting velocities) starting either from an active model and the newly available inactive structure of GHSR. Our new results are described in the new version of the manuscript.

b) Which state does the original model represent? Can the comparison to the recently published structure be more thorough than simply showing a Calpha(?) RMSD (Figure S8)? Are the enhanced sampling MD carried out in presence or absence of ligand and why? Finally the method section as well as the Results section explaining the enhanced sampling protocol should be clarified such that it is not necessary to read the original paper explaining the method to understand.

The initial state of the receptor in the first submission was inactive without any ligand (PDB id: 4GRV). However, we did not use this structure anymore in the current version of the manuscript, as it was a homology model from neurotensin receptor (NTS1R). We decided to start all over from the newly available GHSR inactive structure (this structure was not released when we designed the initial study).

Considering the orthosteric site and the presence of the ligand, only the structure of GHSR bound to a synthetic antagonist is available so far. We previously published GHSR bound to its endogenous agonist ghrelin (19), but it relies only on molecular modeling with a coarse-grained description. The choice not to have any ligand involved is thus based first on the lack of structural information with agonist, and second on the fact that GHSR has a large constitutive activity (above 50%) and can be activated easily without ligand. We hypothesize as well that presence of an agonist/inverse agonist would mainly decrease the energetic barrier associated with conformational changes, which we circumvented by starting from both inactive and active models of GHSR.

c) Many key details of the simulations are missing: number of lipids, number of waters, electrostatics method (as written, it sounds like they didn't use Ewald, which would be a huge problem).

We added more information about systems and simulation setups to the method section of the manuscript. All simulations were run with Ewald summation. We modified the text accordingly: “direct non-bonded interactions were truncated at a distance cut-off of 12Å applying a switching function in the 10:12Å range, while long range electrostatics were computed via Particle Mesh Ewald (PME)”.

d) There is no discussion of statistical convergence. The simulations are very short by today's standards, and the reviewers saw no reason to assume the protein has stopped systematically changing after 300-350 ns (given the uncertainty of starting from a homology model), let alone begun actually sampling. The only evidence is Supp Figure 5, which shows the RMSD is still increasing, while nothing at all is shown for the mutants.

as stated above, we decided to run several long MD simulations for the revised version manuscript, which (we hope) now respects today’s standards (5 replicas of 5 µs for both inactive and active states, for a total of 50 µs). These simulations converged in terms of hydration patterns, as they are very similar in all simulations and moreover specific to each starting state. We also checked the structural divergence from the starting structure through Root Mean Square Deviation (RMSD) calculations (see Author response image 2). We observed expected values with higher RMSD when starting from the activated model (generated by Targeted Molecular Dynamics – TMD) in orange in comparison to simulations starting from the experimental inactive structure in blue.

**Author response image 2. sa2fig2:** Root Mean Square Deviation (RMSD) of all MD simulations with initial structures as references. The 5 MD simulations from the inactive state are represented in blue, and the 5 MD simulations from the active state are represented in orange. Running averages on top of curves are shown for clarity.

e) As it stands, too many details are missing for these calculations to be repeated. Please collect and document all of the scripts used (building the system, running, and analyzing) and put them either into the supplementary info or better yet into a separate repository (e.g. GitHub, zenodo, etc).

We used a classical MD approach for this revised version of the manuscript. We generated our systems using CHARMM-GUI, where all scripts are available on their website. We did not modify these scripts except for the equilibration phase (see Materials and methods section in the revised manuscript).